# A New Species of *Limnephilus* (Insecta: Trichoptera: Limnephilidae) from China, with Revision of the Genus *Limnephilus* on the Chinese Mainland [note 1]

**DOI:** 10.3390/insects13070653

**Published:** 2022-07-19

**Authors:** Haoming Zang, Xinyu Ge, Lang Peng, Changhai Sun, Beixin Wang

**Affiliations:** Lab of Insect Taxonomy & Aquatic Insects, College of Plant Protection, Nanjing Agricultural University, Nanjing 210095, China; 2021102088@stu.njau.edu.cn (H.Z.); 2019202038@njau.edu.cn (X.G.); 2020202045@stu.njau.edu.cn (L.P.)

**Keywords:** species groups, *Limnephilus*, China, morphology, *COI*, immature stages

## Abstract

**Simple Summary:**

As the most diverse genus in the family Limnephilidae, *Limnephilus* Leach, 1815 (194 extant species and 9 fossil species), has a wide distribution and a broad range of ecological tolerances among its species. Although it has a high species diversity in the Holarctic region, only 16 species have been recognized in China until now, of which 7 are endemic to China. In this study, a new species of the genus *Limnephilus* was found. The adult females, larvae, and pupae of this new species were associated with the adult males based on *COI* sequencing, and all the life stages except for the egg stage are described and illustrated. In addition, the biological and habitat information for this new species is provided. Furthermore, all of the *Limnephilus* species from the Chinese mainland are revised at the species group level, and diagnoses for species groups and species (keys of groups and species) are given, along with a distribution map.

**Abstract:**

Fifty individuals of *Limnephilus* from the Qinghai Province, China, were examined, and their *CO**I* barcode sequences were extracted and analyzed. Forty individuals of *Limnephilus* from the Insect Collection of Nanjing Agricultural University (ICNAU), China, were examined, and photos of the male genitalia of four *Limnephilus* species are here presented. The males, females, larvae, and pupae of a new species, *Limnephilus deqianensis* **n. sp.**, associated via *COI* barcode sequences, are described and illustrated. Ecological photos of the male, pupal case, and the habitat of the new species *L. deqianensis* **n. sp.** are also provided. Five species groups containing all seventeen Chinese *Limnephilus* species are revised. Diagnoses, keys, and a distribution map of them are provided. All of the sequences have been uploaded to GenBank. All specimens are deposited in the ICNAU, Nanjing, Jiangsu Province, P. R. China.

## 1. Introduction

Limnephilidae Kolenati, 1848, is one of the most diverse caddis families of the Northern Hemisphere, containing 4 subfamilies and 101 genera [1].

*Limnephilus* Leach, 1815, is the most diverse genus in the family Limnephilidae [2], containing 194 extant species and 9 fossil species [1,3,4]. Out of all known species, 82 are endemic to the Nearctic region; 44 to the West Palearctic region; 20 to the East Palearctic region; 9 to the Oriental region; 3 to the Neotropical region; and 29 of them are distributed in 2 regions, with 16 of them distributed in 3 or more regions.

Schmid [5] established 37 species groups to accommodate all 114 species known at that time, 14 of which were monotypic; 19 species were isolated, and 7 species were treated as *incertae sedis*. However, he did not present the diagnostic characteristics for each group. Afterwards, Nimmo [6,7] reviewed 36 *Limnephilus* species found in Alberta and eastern British Columbia (including his 3 new species, but these were synonymized by various authors later). He followed Schmid’s arrangement, ascribing 33 species from that area to 16 species groups (actually 15 species groups; among them, *L. alvatus* Denning, 1968, was considered to be a synonym of *L. major* Martynov, 1909, from the *L. incisus* Species Group; *L. minusculus*, Banks, 1907, was synonymized as *L. janus* Ross, 1938, and was placed into the *L. incisus* Species Group by Schmid [5], but was considered as a valid species and was transferred into the *L. fenestratus* Species Group by Nimmo [6]; *L. alvatus* Denning, 1968, was placed into the *L. incisus* Species Group by Nimmo, then it was synonymized as *L. major*, Martynov, 1909, by Ruiter in 1995 [8]), and established 3 new monotypic groups for the 3 remaining species. Furthermore, Nimmo presented the diagnoses for these 15 species groups and illustrations for all 36 species. Ruiter [8] organized the Nearctic region species into species groups that were similar to those of Schmid and Nimmo. He also provided diagnostic characteristics for these species’ groups and keys with figures for nearly all New World species. His species group system mostly reflects changes in new data from females and from variation discovered among newly available specimens [9]. However, in recent years, newly published species have rarely been ascribed to species groups by their authors.

Despite the large number of known species in the world, only 16 known species (Table 1) have been recorded in the Chinese mainland [3,10,11]. Among them, 7 are endemic to China [3,12,13,14,15,16]. Moreover, most species of *Limnephilus* were recognized before the 21st century and only 10 new species of *Limnephilus* have been recorded in the last 20 years [3,4,14,15,16,17,18,19].

According to Schmid, among the *Limnephilus* species from the Chinese mainland, *L. fuscovittatus* Matsumura, 1904, and *L. sibiricus* Martynov, 1935, were isolated species, and *L. incertus* Martynov, 1909, and *L. mclachlani* Martynov, 1909, were *incertae sedis.* An amount of 7 out of the remaining 12 species were placed into 5 species groups by Schmid [5], i.e., *L. correptus* McLachlan, 1880, was placed in the *L. flavicornis* Species Group, and *L. politus* McLachlan, 1865, and *L. abstrusus* McLachlan, 1872, were placed in the *L. stigma* Species Group; *L. externus*, Hagen, 1861, was placed in the *L. externus* Species Group; and *L. alaicus*, Martynov, 1915, *L*. *tricalcaratus*, Mosely, 1936, and *L. samoedus*, McLachlan, 1880, were placed in the *L. asiaticus* Species Group; another 5 species that were published after 1955 were not ascribed to the proper species groups by their various authors.

The different members of *Limnephilus* have broad ranges of different ecological tolerance and a strong affinity for lentic habitats [20,21,22]. They inhabit lakes [23,24], permanent or temporary ponds [25,26,27,28], pools [29], and even salt marshes [21]; some species also inhabit lotic habitats such as free-flowing rivers [30], streams, and springs [31]. The genus is particularly well represented at low altitudes and high latitudes, with many subarctic species [32]. Some species, however, are found at high altitudes [19,25,31].

In this study, 50 individuals of *Limnephilus*, collected from the Qinghai Province, China, were examined, and their barcode gene sequences were generated and analyzed. One species, *L. deqianensis* **n. sp.**, was found to be new to the *Limnephilus* fauna; additionally, 5 females, 5 pupae, and 2 larvae were associated with its males by using molecular analysis. We described the larva, pupa, adult male, and female of *L. deqianensis* **n. sp**. We also provide biological information and ecological photos of them. Furthermore, we ascribed 6 other Chinese *Limnephilus* species to 3 species groups; provided diagnoses for 5 species groups and species within 5 species groups, and the keys to 5 species groups and to species within 2 species groups, with a distribution map of all 17 Chinese mainland *Limnephilus* species.

## 2. Materials and Methods

Specimens of the new species studied in this research were collected in Qinghai Province, China. A total of 50 specimens, including 20 adult males and 20 females, 2 larvae, and 8 pupae, were examined in this research. Additional specimens studied in this research were from ICNAU.

### 2.1. Sample Collection

Larvae and pupae were collected by handpicking specimens off of stones in a high-altitude stream (Figure 1C). Only one male adult (Figure 1A) was captured by hand during collection; the other 40 adults were reared from pupae collected at the site. The adult male, larvae and some pupae were preserved in 95% ethanol in the field directly after collection. Other collected pupae were returned to the temporary residence for rearing and the adults that emerged from them were stored in 95% ethanol after they were completely sclerotized about 3–5 days later. Two larvae were found in their pupal cases. We determined that the 2 larvae are the final instar when comparing the metamorphotype larval frontoclypeal apotome remaining in the pupal cases to our reared pupae. Ecological photos of the habitat, the collected male, and pupal case (Figure 1B) were taken with a VIVO X60Pro smartphone. Types and voucher specimens were deposited at the Insect Collection of Nanjing Agricultural University (ICNAU), Nanjing, Jiangsu Province, P. R. China.

### 2.2. Morphological Study

The following methods were used for specimen preparation: abdomens of the adults were cut from their bodies, transferred into 10% NaOH solution, and heated to 90 °C for 2 h to remove all of the non-chitinous tissues. Then, the cleaned abdomens were rinsed in distilled water and mounted on a depression slide with 75% ethanol for examination. Photographs of the male and female genitalia, larvae, and pupae were taken by using a stereomicroscope (Nikon SMZ18), and photos of the female spermathecal structure were taken by using a compound microscope (Olympus BX41). Each of these pieces of equipment had a built-in camera and Nis-Element D^®^ software (Version 3.22.14, Nikon, Shanghai, China). A series of photos at different focal depths were taken and stitched together with Zerene Stacker^®^ software (Zerene Systems LLC, Richland, WA, USA) and then arranged using Adobe Photoshop^®^ (version 2017.0.0, Adobe, San Jose, CA, USA). Subsequently, each abdomen was stored in a microvial together with the remainder of the specimen in 75% ethanol.

The terminology for the male genitalia follows that presented by Schmid [32]; the terminology for the female genitalia was adapted from that of Schmid [32], and for the female spermatheca, the terminology was adapted from that of Nielson [33]. The terminology for the larva corresponds with that of Wiggins [34], while for the pupae, it conforms to that of Denning [35].

### 2.3. Molecular Analysis

The left hind legs of 12 individuals (males, females, and larvae) and left forelegs of 5 individuals (pupae) were taken from the bodies for DNA extractions. Extraction followed the animal tissue protocols of the Ezup Column Animal Genomic DNA Purification kit (Sangon Biotech, Shanghai, China). The PCR amplification, fragment sequencing, and analyses followed the procedures of Xu [36]. The primers [37] (LCO1490/HCO2198) are listed in Table 2. Neighbor-joining (NJ) trees were constructed using Mega v10.2.6 [38]. Calculation parameters were set as follows: Kimura 2-parameter substitution model, pairwise gap deletion and others as defaults. The NJ tree was plugged into ITOL v5 [39]. Female, larval, and pupal specimens were associated with adult males for *L. deqianensis* **n. sp.** according to the criteria outlined by Zhou [40]. Their *COI* sequences were uploaded to GenBank and their accession numbers are included in Table 3.

## 3. Results

*Limnephilus deqianensis* Zang & Sun, **n. sp.** (Figure 1, Figure 2, Figure 3, Figure 4, Figure 5, Figure 6, Figure 7, Figure 8, Figure 9, Figure 10, Figure 11, Figure 12, Figure 13 and Figure 14)

*Holotype male*: CHINA, Qinghai Province, Men-yuan County, De-qian Town, 37°21′83″ N, 102°20′37″ E, alt. 2930.0 m, 2 July 2021, from a pupa hand-picked from a stone in the stream and raised indoors, Zang Hao-ming. Paratypes: same data as holotype, 19 males, 20 females, 2 larvae, and 8 pupae.

*Etymology*: The specific epithet is from the Chinese name of the type locality, De-qian Town, Qinghai Province.

*Distributions*: China (Qinghai).

### 3.1. Description

*Adult*: Body brown, 9.5–10.1 mm long for males (*n* = 20), and 10.8–11.9 mm long for females (*n* = 20). Head brown, antennae brown, 13.6–13.6 mm long for males (*n* = 20) and 14.3–14.8 mm for females (*n* = 20). Forewings (Figure 2A) 14.0–14.2 mm long for males (*n* = 20) and 14.7–15.8 mm long for females (*n* = 20), brown, densely covered with dark spots, each with dark stigma, huge pale spot near apex of Rs cell and thyridial cell, small spot in each base of fork II and III. Hind wings wider than forewings, transparent, without any dark brown spots. Spur formula 1-3-4; forefemora of male, each with lower margin having black setal brush (Figure 2B).

Male genitalia. Tergum VIII in dorsal view (Figure 3A) with basal portion subrectangular and distal portion triangular and produced into a bifid apicodorsal lobe; in lateral view (Figure 3B), basal portion subrectangular, apicodorsal lobe with apex beak-shaped. Segment IX in dorsal view (Figure 3C), with anterior margin straight and posterolateral corners each produced into small process directed posteromesad (Figure 3F); in lateral view (Figure 3D and Figure 4A), with anterior margin convex anteriad, upper portion triangular, lower portion rhomboid and narrowing ventrally. Superior appendages flaplike, oval in lateral view, spatulate in dorsal view (Figure 3C,D and Figure 4A–C). Intermediate appendages each divided into basal plate and apical lobe; basal plates in lateral view (Figure 3D and Figure 4A), each somewhat triangular, with lower portion of each inner margin produced into small membranous process, in caudal view (Figure 3F and Figure 4C), each somewhat semicircular; apical lobes in lateral view (Figure 3D and Figure 4A), each tapered to apex and with apex slightly curved upwards; in dorsal view (Figure 3C), each digitate. Inferior appendages fused with segment IX in lower portion of posterior margins; in lateral view (Figure 3D and Figure 4A), higher than long, each with upper portion thicker than lower portion, upper portion with posterior margins sinuate, and produced into the free portion. Phallic apparatus slender, phallocrypt cylindrical tube (Figure 3C,D), endotheca membranous. Aedeagus (Figure 3C–E) slender, tube-like, tapering from base to apex, slightly curved upwards. Parameres (Figure 3D,G and Figure 4D) each slightly longer than aedeagus, in lateral view tubular with flattened setose apex, and having a small setose lobe on inner side.

Female genitalia. Tergum VIII subrectangular in dorsal and lateral views (Figure 5A,B). Sternum VIII in lateral view (Figure 5B), subrectangular. Segment IX in dorsal view (Figure 5A and Figure 6B), trapezoid, in lateral view (Figure 5B and Figure 6A), somewhat subrectangular, with paired button-like processus. Segment X fused with segment IX, tubular in both caudal and lateral views (Figure 5B,C and Figure 6A,D), with upper portion deeply incised anteriad. Vulvar scales (Figure 5F and Figure 6C), with median vulvar lobe semicircular and paired lateral vulvar lobes each triangular. Spermatheca (Figure 5E,G and Figure 6E,F) somewhat elliptic, spermathecal sclerites slightly sclerotized, each with lateral margin arc-shaped, with apodeme straight and strongly scerotized, somewhat calvate; processus spermatheca obviously divided into 2 sections, anterior section slightly sclerotized, saccular in lateral view (Figure 5G and Figure 6E), and vase-like in dorsal view (Figure 5E); posterior section membranous, elliptic in lateral view (Figure 5G) and somewhat circular in dorsal view (Figure 5G).

*Larva (final instar)*: Body length 15.7–16.5 mm (*n* = 2). Head, pronotum and mesonotum strongly sclerotized.

Head. Head capsule nearly ovoid, dark brown with irregular yellow patches (Figure 7A–D), 2.4–2.6 mm long and 1.7–1.9 mm wide in dorsal view (*n* = 2). Some muscle scars longitudinally arranged on posteromedial portion of frontoclypeal apotome (Figure 7A,C) and randomly arranged on each of parietal apotome (Figure 7A,C). Frontoclypeal suture (Figure 7A,C) Ʊ-shaped with frontoclypeal apotome posterior angle in about 100°, each frontoclypeal suture at seta 5 curved at about 120°. In lateral view (Figure 7E), eyes oval, white (color probably due to speciemens being prepupae, having already begun to molt into pupae). Some muscle scars randomly arranged on each side from postgena to parietal. In ventral view (Figure 7B), anterior ventral apotome (avp) narrowly subpentagonal, with anterior margin dark brown; ventral ecdysial line on posterior meson about half as long as anterior ventral apotome, posterior ventral apotome absent. Pairs of primary setae (Figure 7B–D) 1–7, 9–17 arising on dorsal surface of head; setae 8 and 18 arising on ventral surface of the head. Setae 1, 4, 10, and 16 golden, hair-like; setae 10 thinner and shorter than 1 and 16; other setae black, spike-like; seta 14 longest and thickest; setae 2, 3, 5, 7, and 17 short and thick; setae 15 long and slender; setae 6 and 11 short and fine; setae 8, 9, 12, and 13 of medium size, setae 18 broken away, leaving only alveoli. Labrum (Figure 7A,C) brown, with middle portion of anterior margin slightly concave, and a setal brush at each anterolateral corner.

Mandibles (Figure 8A,C) shaped for scraping, black, asymmetrical in shape and arrangement of hairs; only left mandible with stiff hairs on subbasomesal edge of dorsal margin. Cardo (Figure 7B) subpentagonal, with inner and outer ends black, and intermediate portion yellow to brown.

Thorax. Pronotum (Figure 9A) subrectangular, brown in dorsal view, subdivided longitudinally by mesal suture, with several black, spike-like setae; shallow transverse depression about 1/3 distance from anterior margin; posterior ridge dark brown; Prosternal horn (Figure 9D) present and apical half sclerotized. Prosternite (Figure 9D) subpentagonal, convex anteriorly, slender laterally, and posteromesal margin slightly concave. Paired lateral posterior prosternites (Figure 9D) with irregular shape.

Mesonotum (Figure 9B) subrectangular dorsally, subdivided longitudinally by mesal suture, each part yellowish brown, with scattered long spike-like setae and random muscle scars; each anterior margin with convex black spot mesal of anterolateral corner, dark spots forming transverse W-shaped mark across middle of pair of nota, posterolateral margins each black and produced ventrolaterad as small triangular projection, posterior margins black except for pair of narrow yellowish brown transverse bands mesal of posterolateral corners.

Metanotum (Figure 9C) with *sa1* sclerites large, elongate triangular, close to each other, each having 4–6 setae; *sa2* sclerites each large, subrectangular-oval, widely separated, with 2–3 pairs of small rounded sclerites mesally between them. *Sa3* sclerites each subrectangular, with mesal margin irregular, with some scattered, strong, short and long setae.

Legs. Legs yellowish to light brown. Forelegs (Figure 10A,B) shorter and slightly thicker than mid- and hind-legs. Forecoxae (Figure 10A,B) shorter than mid- and hind- coxae and conical in lateral view, each anterior side (Figure 10B) with 10–12 black spike-like setae. Foretrochanters (Figure 10A,B) each 2-segmented, both segments subtriangular, basal joint slightly shorter than apical one; apical joint with anterior side (Figure 10B) bearing 3 light-brown spur-like major setae. Forefemora (Figure 10A,B) each cylindrical with upper margin arc-shaped, with 6 brown spike-like seta dorsally; lower margin straight, with 2–3 brown spike-like setae; anterior side (Figure 10B) with 3 short black setae mesally. Foretibiae (Figure 10A,B) each cylindrical, slightly curved, with apex slightly enlarged, with 2 light brown apical spurs on anterior side and ventral margin. Foretarsi cylindrical, without setae. Foretarsal claws (Figure 10A,B) stouter and shorter than on mid- and hind legs, somewhat conical, brown and slightly curved downwards.

Mid- and hind-coxae (Figure 10C–F) longer and slenderer, each bearing a row of black spike-like setae on upper margins. Mid- and hind-trochanters (Figure 10C–F) each 2-segmented with basal joint subtriangular and shorter than apical joint; apical joint trapezoidal with anterior side (Figure 10D,E) bearing 3 light brown spur-like major setae, with ventral margin having trochanteral brush. Each mid-femur (Figure 10C,D) with anterior side (Figure 10D) bearing 4 black short setae and posterior side (Figure 10C) 2 setae. Each hind femur (Figure 10E,F) with upper margin bearing 5 brown spike-like setae and with anterior side (Figure 10E) having 6 short black short setae and posterior side (Figure 10F) having 5.

Light brown major femoral setae present in all legs. All femora with lower margins bearing inconspicuous setal buds. Mid- and hind-tibiae (Figure 10C–F) are same shape as foretibiae, but slightly more elongate; tibial spurs present same as foretibiae. Light brown spur-like basal setae present in all legs.

Abdomen. Abdominal segment I (Figure 11A,B) with 1 dorsal and 2 lateral humps. About 60 black setae arranged in two rows on tergum I (Figure 11A); anterior row with about 40 setae and most of them each with basal sclerite; posterior row with about 20 setae and most with basal sclerites smaller than in anterior row, only 2 of them same size as anterior sclerites. Sternum I (Figure 11B) with 2 triangular median *sa2* sclerites and 2 irregular lateral *sa3* sclerites; each median sclerite with about 20 setae and lateral sclerites each with about 9 setae.

Tracheal gills trifid, bifid or unbranched. Gill forms and their distributions from segments II–VII as in Table 4. Lateral fringe present on each side from segment II to segment VIII.

Dorsal sclerite of segment IX (Figure 11C) semicircular and yellowish, with about 10 long black spike-like setae arranged in arc, intermingled with 5 short fine black setae. Lateral sclerites (Figure 11C,D) each with 7 long, dark spike-like setae apically, 1 long black seta subapically, and 1 long seta and about 3 short black setae mesally. Ventral sole plate (Figure 11D) triangular with anterior margin black in lateral view. Anal proleg claws (Figure 11C,D) brown and strongly sclerotized, each with about 6 fine yellowish basal setae, dorsal accessory hook present, shorter and finer than main claw.

*Pupa*: Length 13.7–14.6 mm (*n* = 10). Antennae slightly shorter than body, each straight at the end; scape longer and thicker than remaining joints. Each eye with entire posterior margin having 17–19 curved hair-like setae (Figure 12C) overlying eye surface. Labrum (Figure 12B) wide, brown with 14 strong and slender setae on surface. Mandibles (Figure 12C) blackish brown and crossing each other apically, distal parts relatively slender; median edges of blades slightly serrated; basal parts yellowish and each with 2 strong black hair-like setae. Wing sheaths reaching end of abdominal segment IV. Mid- and hind-legs natatorial; tarsal segments I–IV (Figure 12E,G) each with dark fringe. Abdominal segments V–VIII fringed (Figure 12G).

Anterior pair of dorsal hook plates on abdominal terga III–VII, each with 4–6 strong and curved hooks. IIIa (Figure 13B) relatively small, suboval; IVa (Figure 13C), Va (Figure 13D), VIa (Figure 13E) and VIIa (Figure 13F) round and subequal in size, each with strong and fine teeth more strongly sclerotized than supporting hook plate.

Posterior pair of dorsal hook plates on abdominal terga I and V. Ip (Figure 13A) with each anterior margin indiscernible, with stubby teeth arranged mesally. Vp (Figure 13D) subrectangular with 21–23 strong and short hooks more strongly sclerotized than supporting hook plate. Hook numbers on each hook plate as follows: Ip 13–15; IIIa 4; IVa 3–4; Va 4–5; Vp 21–23; VIa 4–5, VIIa 4–5. Gill forms and their distributions from segments II–VII are list as in Table 5.

Anal appendages (Figure 12D) long and apically sclerotized, in dorsal and ventral views each inner margin bearing row of black spike-like setae; apex bent outward and concave laterally.

### 3.2. Diagnosis

Male: The male genitalia of this new species are similar to those of *Limnephilus kaumarajiva* Schmid, 1961, from Pakistan, but differ from those of the latter in that (1) the superior appendages are small and slightly shorter than the intermediate appendages in the lateral view (large and significantly longer than the intermediate appendages in lateral view in *L. kaumarajiva*); (2) the intermediate appendages each have their basal plate semicircular in the caudal view (subtriangular in caudal view in *L. kaumarajiva*); (3) the inferior appendages have the apices each bifid (the apices are acute in *L. kaumarajiva*); (4) the parameres each have the single setose subapex flattened and with 1 small setose subapical lobe on its inner side (the parameres each have 2 flattened setose apices and 1 small setose subapical lobe on its inner side in *L. kaumarajiva*).

Female: The female genitalia of this new species are similar to those of *Limnephilus primoryensis* Nimmo, 1995, from Russia (Primorye). This female was described by Nimmo, 1995 [41], but *L. deqianensis* differs from the latter in that (1) segment IX is subtriangular and broad in the middle in the lateral view (the upper portion of the posterior margin is concave anteriorly, narrow in the upper portion in *L. primoryensis*); (2) segment IX has paired button-like processus (this processus is absent in *L. primoryensis*); (3) segment X is narrow in the lateral view (segment X is broad and triangular laterally in *L. primoryensis*).

Larva: The larva of this new species is similar to that of *Limnephilus ademus* Ross, 1941, from America; this larva was described by Ruiter [42], but differs from the latter in that (1) the head capsule is without blotches outside the anterior constriction of the frontoclypeal suture (with two blotches primarily outside the anterior constriction of the frontoclypeal suture in *L. ademus*); (2) the meso and metafemora are without accessory setae on the lateral margins in the new species (accessory setae are present on the lateral margins of the meso and metafemora in *L. ademus*); (3) 5 small brown sclerites are arranged randomly between the metanotal *sa2* sclerites (without such 5 small sclerites in this position in *L. ademus*).

Pupa: The pupa of the new species is far from any known described pupa and can be distinguished from other species by the numbers of hooks on each hook plate.

### 3.3. Biology and Habitat

*Limnephilus deqianensis* **n. sp.** larvae and pupae were found in a small high-altitude stream (2930 m) in open areas. The riparian vegetation is mainly alpine meadow. Stream margins were covered with detritus (mainly grass blades). The water in the stream was clear and transparent at the collection site. The anterior ends of each pupal case was attached under a flat stone at the bottom of the stream. The species has the habit of aggregating to pupate. A newly emerged male was found during collection, and the pupae in the pupal cases were well developed.

### 3.4. Molecular Analysis

Nineteen mitochondrial *COI* sequences, including seventeen from *Limnephilus deqianensis*
**n. sp.**, and two downloaded from GenBank were pruned to 658bp to construct the NJ tree (Figure 14).

The genetic distances are displayed at the node and rounded to four decimal places. Larvae, pupae, and female adults of *Limnephilus deqianensis* **n. sp.** were clustered with the male adults of *L. deqianensis* **n. sp.** The maximum genetic distances of the *L. deqianensis* **n. sp.** cluster were lower than 2.43%. Therefore, we concluded that the association is successful.

### 3.5. Revision of Limnephilus Species from the Chinese Mainland

Key to the males of 5 species groups of *Limnephilus* species occurring on the Chinese mainland:

1. Superior appendages, each with posterior margin having incision (as figures on p. 34 in Malicky, 2011)…………
*……………………………………………………………………………………………L*. *perpusillus* Species Group *nomen novum*-Superior appendages without incision……………………………………………………………………………………**2**2.Superior appendages, each triangular with apical corner acute and inferior appendages, each with free portion bearing an apicodorsal angle produced upwards in lateral view (as Figure 320 in Nimmo, 1971)………………… 
*………………………………………………………………………………………………………………...L*. *externus* Species Group-Superior appendages, each trapezoidal, pentagonal, ligulate, or irregular, free portion of each inferior appendage with apicodorsal angle not produced upwards in lateral view…………………………………………………………**3**3.Superior appendages trapezoidal and intermediate appendages straight in lateral view, setose portion of each paramere about half as long as whole length (Figure 15)…………………………………*L*. *flavicornis* Species Group-Superior appendages ligulate, pentagonal, or irregular, intermediate appendages curved obviously in lateral view, setose portion of each parameres shorter than half of whole length…..…………………………………………**4**4.Superior appendages shorter than 1/2 height of segment IX in lateral view (Figure 3 and Figure 4)……………………………
*………………………………………………………………………………………………………………...L*. *asiaticus* Species Group-Superior appendages longer than 1/2 height of segment IX in lateral view (Figures 17 and 18)………………………
*…………………………………………………………………………………………………………………...L*. *stigma* Species Group

#### 3.5.1. *Limnephilus*
*perpusillus* Species Group *nomen novum* (*L**. incisus* Species Group of Schmid, 1955)

We proposed renaming this species group to the “*L*. *perpusillus* Species Group”, from the species *L. incisus* Curtis, 1834, which was used to name the group when it had been transferred to the genus *Colpotaulius* by Vshivkova et al. [43], and *L. perpusillus* Walker, 1852, which was the oldest species in the group. Originally, this species group included the following 7 species: *L. hyalinus* Hagen, 1861; *L. acnestus* Ross, 1938; *L. ademus* Ross, 1941; *L. janus* Ross, 1938; *L. major* Martynov, 1909; *L. perpusillus* Walker, 1852; and *L. secludens* Banks, 1914. The species *L. homeros* Malicky, 2011, was distributed in China and was not ascribed to any species group by the author [16]. Here, we place *L. homeros* Malicky, 2011, into this species group based on its male genitalia characteristics. The group can be diagnosed by following characteristics; (1) superior appendages, each with a posterior margin having an incision, as in *L. major* Martynov, 1909, *L. secludens* Banks, 1914, *L. ademus* Ross, 1941, *L. acnestus* Ross, 1938, and *L. homeros* Malicky, 2011; or subtriangular and curved and extended toward intermediate appendages in the caudal view as in *L. hyalinus* Hagen, 1861, *L. janus* Ross, 1938, and *L. perpusillus* Walker, 1852 (Table 6); (2) inferior appendages, each with the fused portion narrowed, as in *L. major* Martynov, 1909, *L. secludens* Banks, 1914, *L. homeros* Malicky, 2011, *L. hyalinus* Hagen, 1861, *L. ademus* Ross, 1941 and *L. acnestus* Ross, 1938; or almost invisible, as in. *L. janus* Ross, 1938, and *L. perpusillus* Walker, 1852 (Table 6); (3) aedeagus with a thick extensible extremity.

*Limnephilus homeros* Malicky, 2011

*Limnephilus homeros* Malicky, 2011, 35, figure p34, male, China.

*Diagnosis*: (1) Superior appendages, each with the posterior margin having a strong incision; (2) intermediate appendages are subrectangular, with the upper and lateral margins concave in the caudal view; (3) inferior appendages, each with the free portion forming an apicodorsal lobe and with the fused portion broader than the free portion; (4) parameres slightly longer than the aedeagus, each with setose lobe subapically.

*Type country*: China.

*Distribution*: China (Sichuan).

#### 3.5.2. *Limnephilus*
*externus* Species Group of Schmid, 1955

This species group contains the following two species: *L. externus* Hagen, 1861, and *L. thorus* Ross, 1938, of which *L. externus* Hagen, 1861, was distributed in China. The group can be diagnosed by following characteristics: (1) superior appendages are triangular, each with the upper margin straight and the apical corner acute in the lateral view; (2) inferior appendages, each with the free portion bearing an apicodorsal angle produced upwards; (3) parameres, each with flattened setose subapex, and each setose portion shorter than 1/2 of the whole length.

*Limnephilus externus* Hagen, 1861

*Limnophilus externus* Hagen, 1861, 257, female, Canada. 477.

*Limnophilus externus* Tjeder, 1930, 201–203, Figure 1, Figure 2, Figure 3, Figure 4, Figure 5, Figure 6, Figure 7 and Figure 8.

*Limnophilus congener* McLachlan, 1875, 56–57, pl. 8, Figure 1, Figure 2, Figure 3, Figure 4, Figure 5, Figure 6, Figure 7 and Figure 8, Russia, Finland. 479 (synonymized by Ulmer 1907 [44]).

*Limnephilus luteolus* Banks, 1899, 207–208. 480 (synonymized by Ross 1944 [45]).

*Limnephilus oslari* Banks, 1907, 121–122, pl. 9, Figure 19. 481 (synonymized by Milne 1935 [46]).

*Limnephilus tersus* Betten, 1934, 334, pl. 46, Figure 6, Figure 7 and Figure 8, pl. 47, Figure 1, Figure 2, Figure 3, Figure 4 and Figure 5 (synonymized by Ross 1944 [45]).

*Diagnosis:* (1) Superior appendages are triangular, each with the inner side of posterior margin bearing a row of black teeth; (2) intermediate appendages are curved downward in the lateral view; (3) inferior appendages, each with the free portion bearing an apicodorsal angle produced upwards; (4) parameres, each with the apicodorsal margin having a row of fringe directed anterad.

*Type country:* Canada.

*Distribution:* China (Shanghai); America; Canada; Finland; Norway; Russia; Sweden.

#### 3.5.3. *Limnephilus*
*flavicornis* Species Group of Schmid, 1955

This species group includes the following 3 species: *L. correptus* McLachlan, 1880 (East Palearctic and Oriental, including China), *L. ectus* Ross, 1941 (Nearctic), and *L. flavicornis* (Fabricius, 1787) (East and West Palearctic). The group can be diagnosed by the following characteristics: (1) superior appendages are trapezoidal; (2) posteromedial portion of each superior appendage bears an arc of black teeth on the inner side; (3) intermediate appendages are slender and straight, and are as long as the superior appendages, as in *L. correptus* McLachlan, 1880, and *L. flavicornis* Fabricius, 1787, or slightly shorter than superior appendages, as in *L. ectus* Ross, 1941; (4) setose portion of each paramere is about half as long as the whole length.

*Limnephilus correptus* McLachlan, 1880 (Figure 15)

*Limnophilus correptus* McLachlan, 1880, 18, pl. 53, Figure 1 and Figure 2, Russia.

*Diagnosis*: (1) Superior appendages are trapezoidal, with the posteromedial portion of the inner side bearing an arc of black teeth; (2) intermediate appendages are straight and slightly longer than the superior appendages; (3) parameres, each with a trapezoidal setose apex and a stubby setose lobe in the lateral view.

*Type country*: Russia.

*Distribution*: China (Sichuan, Heilongjiang); Japan; Mongolia; North Korea; Russia.

*Material examined*: CHINA, Hei Long-jiang Province, Hei-he City, Wudalianchi County, 8 August 1987, leg. Xue Yin-gen.

#### 3.5.4. *Limnephilus asiaticus* Species Group of Schmid, 1955

Originally, this species group included the following 11 species: *L. alaicus* Martynov, 1915, (East and West Palearctic, including China); *L. asiaticus* McLachlan, 1874, (East and West Palearctic and Oriental); *L. centralis* Curtis, 1834, (West Palearctic); *L. chereshnevi* Nimmo, 1995, (East Palearctic), *L. frijole* Ross, 1944 (Nearctic); *L. hirsutus* Pictet, 1834, (East and West Palearctic); *L. labus* Ross, 1941, (Nearctic); *L. pallens* Banks, 1920, (Nearctic); *L. quadratus* Martynov, 1914, (East and West Palearctic); *L. samoedus* McLachlan, 1880, (East Palearctic, Nearctic, and Oriental, including China); and *L. tricalcaratus* Mosely, 1936, (East Palearctic and Oriental, including China). The isolated species in Schmid’s system, *L. fuscovittatus* Matsumura, 1904; the new species described in this article, *L. deqianensis*
**n. sp.**; *L. mandibullus* Yang & Yang, 2005, and *L. nipponicus* Schmid, 1964, published after 1955, show the diagnostic characteristics of the *L*. *asiaticus* Species Group, so we ascribe them to this species group. The group can be diagnosed by following characteristics: (1) Superior appendages, each ligulate and shorter than half of the height of segment IX in the lateral view; (2) intermediate appendages, each with a braod basal portion.

Key to the males of seven species of the *L*. *asiaticus* Species Group from Chinese mainland:

1Parameres obviously bifid……………………………………………………………………………………*L. mandibullus*-Parameres unbranched or having only small setose lobe………………………………………………………………..**2**2Parameres with flattened apices………………………………………………………………………………*L. deqianensis*-Parameres with apices not flattened………………………………………………………………………………………..**3**3Aedeagus with apical hook longer than extensible extremity……………………………………………………………**4**-Aedeagus without apical hook or with apical hook shorter than extensible extremity………………………………**5**4Intermediate appendages with basal portion as wide as superior appendages………………………………*L. alaicus*-Intermediate appendages with basal portion wider than superior appendages……………………….*L. tricalcaratus*5Superior appendages shorter than 1/3 height of segment IX in lateral view……………………………….*L. samoedus*-Superior appendages longer than 1/3 height of segment IX in lateral view……………………………………………**5**6Inferior appendages each with narrow fused portion and stout free portion……………………………*L. nipponicus*-Inferior appendages each with broad fused portion and stout free portion…….………………………*L. fuscovittatus*

*Limnephilus mandibulus* Yang & Yang, 2005

*Limnephilus mandibulus* Yang & Yang, 2005, 493, Figure 2, male, female, China.

*Diagnosis*: (1) Segment IX is trapezoidal in the lateral view; (2) superior appendages curved downwards; (3) inferior appendages are subtriangular with the apical corner acute; (4) parameres are bifid and setose.

*Type country*: China.

*Distribution*: China (Shaanxi, Gansu).

*Limnephilus deqianensis* Zang & Sun, **n. sp.** (Figure 3 and Figure 4)

*Diagnosis*: (1) Distal portion of tergum VIII is triangular and produces into bifid apicodorsal lobe in the dorsal view; (2) intermediate appendages, each divided into the basal plate and apical lobe; (3) parameres, each tubular with a flattened, setose apex, and having a small setose lobe on the inner side in the lateral view.

*Type country*: China.

*Distribution*: China (Qinghai)

*Limnephilus alaicus* (Martynov, 1915)

*Astratus alaicus* Martynov, 1915, 417–421, Figure 12, Figure 13, Figure 14, Figure 15, Figure 16, Figure 17, Figure 18 and Figure 19, Tajikistan.

*Diagnosis*: (1) Aedeagus long and thick with the extensible extremity bearing an external hook; (2) parameres, each slender, with triplet apices in the dorsal view.

*Type country*: Tajikistan.

*Distribution*: China (Qinghai); Kazakhstan; Mongolia; Russia; Turkey.

*Limnephilus tricalcaratus* (Mosely, 1936)

*Astratus tricalcaratus* Mosely, 1936, 453–454, pl. III, Figure 1, Figure 2, Figure 3, Figure 4, Figure 5, Figure 6, Figure 7 and Figure 8, Western Tibet.

*Limnephilus tricalcaratus* (Mosely, 1936), Schmid, 1955: 140.

*Limnephilus tricalcaratus* (Mosely, 1936), Grigorenko 2002: 110, synonymized with *Limnephilus samoedus* (McLachlan, 1880).

*Limnephilus tricalcaratus* (Mosely, 1936), Oláh, 2019, 36–37, Figures 109–111, China (Tibet), resurrected from synonymy, as *Limnephilus tricalcaratus* (Mosely, 1936).

*Diagnosis*: (1) Aedeagus long and thick with extensible extremity bearing an external hook; (2) parameres, each slender, glabrous, and tapered to an apex and bearing 3 additional spines on the apex; (3) Intermediate appendages, with the basal portion wider than the superior appendages.

*Type country*: China.

*Distribution*: China (Tibet).

*Limnephilus samoedus* (McLachlan, 1880)

*Astratus samoedus* McLachlan, 1880, 16, pl. 53, Figure 1, Figure 2, Figure 3, Figure 4, Figure 5, Figure 6, Figure 7, Figure 8 and Figure 9, Russia.

Previously, Schmid placed this species in the *L. asaticus* Species Group. However, based on the morphological phylogenetic analysis, Vshivkova and her co-authors showed that the species belonged to the genus incertae sedis, not Limnephilus [43]. Studies conducted after that (e.g., Oláh et al. 2019) still treat it as a member of Limnephilus; therefore, we here also place it in the *L. asiaticus* Species Group.

*Diagnosis*: The male genitalia of this species are similar to those of *L. alaicus* and *L. tricalcaratus* but differ from *L. alaicus* and *L. tricalcaratus* in that the apical hook on the aedeagus is shorter than the extensible extremity.

*Type country*: Russia.

*Distribution*: China (Tibet); America; Mongolia; Russia.

*Limnephilus nipponicus* Schmid, 1964

*Limnephilus nipponicus* Schmid, 1964, 834–836, Figures 34–38, male, female, Japan.

*Diagnoses*: (1) Superior appendages are subtriangular in the lateral view; (2) intermediate appendages curve upwards with the basal portion broad, and slightly longer than the superior appendages in the lateral view; (3) inferior appendages, each with the fused portion narrow; (4) parameres, each slightly curved and tapered to the apex with the apical portion setose.

*Type country*: Japan.

*Distribution*: China (Qinghai); Japan; Russia.

*Limnephilus fuscovittatus* Matsumura, 1904 (Figure 16)

*Limnophilus fuscovittatus* Matsumura, 1904, 171, pl. 12, Figure 13, Japan.

Limnophilus subfuscus Ulmer, 1907, 20–21, Figures 32–35, Japan; synonymized by Nakahara (1914) and Nozaki and Tanida (1996).

*Diagnosis*: (1) The male genitalia of this species are similar to those of *L. nipponicus* but differ from the latter in that: (1) the intermediate appendages slightly curved downward and shorter than the superior appendages in the lateral view (curved upward and slightly longer than the superior appendages in the lateral view in *L. nipponicus*); (2) inferior appendages with the fused portion broad (with the fused portion narrow in *L. nipponicus*); (3) parameres, each with the apical portion slightly bulging (tapered to the apical portion in *L. nipponicus*).

*Type country*: Japan.

*Distribution*: China (Liaoning, Sichuan); Japan; Mongolia; North Korea; Russia.

*Material examined*: One male, CHINA, Liaoning Province, Shen-yang City, 25 May 1957.

#### 3.5.5. Limnephilus stigma Species Group of Schmid, 1955

Originally, this species group included the following 7 species: *L. abstrusus* McLachlan, 1872 (East Palearctic, including China); *L. ademiensis* Martynov, 1914 (East Palearctic); *L. flavospinosus* Stein, 1874 (West Palearctic); *L. indivisus* Walker, 1852 (Nearctic); *L. infernalis* Banks, 1914 (Nearctic); *L. politus* McLachlan, 1865 (East and West Palearctic, including China); and *L. stigma* Curtis, 1834 (Nearctic, East and West Palearctic). Limnephilus zhejiangensis Leng & Yang, 2004, and *L. distinctus* Tian & Yang, 1992, first described after 1955, shared common characteristics with other members of this species group. We here ascribed them to this group based on male genitalia characteristics. The group can be diagnosed by following characteristics: (1) tergum VIII projected well posterad from membranous connection to segment IX; (2) superior appendages, each longer than 1/2 the height of segment IX in the lateral view; (3) intermediate appendages long and slender except in *L. stigma* Curtis, 1834; (4) inferior appendages, each with a narrow-fused portion and a slender free portion.

Key to the males of four species of the *L. stigma* Species Group from China

1.Intermediate appendages each with apical portion curved slightly upwards in lateral view………*L. zhejiangensis*-Intermediate appendages obviously curved upwards from middle portion to apex in lateral view………………**2**2.Inferior appendages each with fused portion very narrow and almost invisible………………………*L. distinctus*-Inferior appendages each with fused portion narrow but conspicuous ………………………………………………**4**3.Segment IX posterior margin straight at insertion of inferior appendages……………………………….*L. abstrusus*-Segment IX posterior margin concave at insertion of inferior appendages…………………………………*L. politus*

*Limnephilus zhejiangensis* Leng & Yang, 2004 (Figure 17)

*Limnephilus zhejiangensis* Leng & Yang, 2004, 520, Figures 22–24, male, female, China.

*Diagnosis*: (1) Tergum VIII with a rounded posterior lobe; (2) superior appendages, each subtriangular, with the inner side bearing two rows of teeth; (3) intermediate appendages slightly shorter than the superior appendages, each with the apical portion curved slightly upward; (3) inferior appendages, each digitate with the fused portion narrow; (4) parameres, each with a slightly flattened setose apex and a small subapical setose lobe.

*Type country*: China.

*Distribution*: China (Zhejiang).

*Material examined*: One Paratype male. CHINA, Zhejiang Province, Qing-yuan County, Baishanzu, alt. 1300.0m, 25 October 1993, by light trap, det. Wu Hong.

*Limnephilus distinctus* Tian & Yang, 1992 (Figure 18)

*Limnephilus distinctus* Tian & Yang, 1992, 880, Figure 9, male, female, China.

*Diagnosis*: (1) Superior appendages, each subtriangular with the inner side bearing two rows of teeth; (2) intermediate appendages, each curved strongly upward, and as long as the superior appendages; (3) inferior appendages digitate, each with the fused portion very narrow and almost indiscernible; (4) parameres, each with a subtriangular-flattened setose apex.

*Type country*: China.

*Distribution*: China (Sichuan).

*Materal examed*: One male, CHINA, Sichuan Province, Jiuzhaigou County, 23 August 1994, leg. Du Yu-zhou.

*Limnephilus abstrusus* McLachlan, 1872

*Limnophilus abstrusus* McLachlan, 1872, 62–63, pl. 1, Figure 13, Russia.

*Diagnosis*: (1) Superior appendages subtrapezoidal in the lateral view; (2) intermediate appendages curve upwards and are slightly shorter than superior appendages; (3) posterior margins of segment IX are straight at the insertion of the inferior appendages.

*Type country*: Russia.

*Distribution*: China (Sichuan); Mongolia; Russia.

*Limnephilus politus* McLachlan, 1865

*Limnophilus politus* McLachlan, 1865, 39, pl. 9, Figure 24, Britain.

*Goniotaulius concentricus* Kolenati, 1848

*Diagnosis*: The male genitalia of this species are similar to those of *L. abstrusus* but differ from the latter in that: (1) The length of the free portion of each inferior appendage is about 3/4 of the width of segment IX (inferior appendages, each with the free portion almost as long as segment IX in *L. abstrusus*); (2) posterior margins of segment IX are concave at the insertion of the inferior appendages (straight in *L. abstrusus*).

*Type country*: Britain.

*Distribution*: China (Inner Mongolia); Britain; Czech Republic; Finland; Germany; Hungary; Kazakhstan; Netherlands; Norway; Poland; Russia; Sweden; Ukraine.

#### 3.5.6. *Limnephilus* Isolated Species

*Limnephilus sibiricus* Martynov, 1929

*Limnophilus subfuscus sibiricus* Martynov, 1929, 305–308, Figure 13, Figure 14, Figure 15 and Figure 16, female, China.

*Limnephilus spurisi* Grigorenko, 2002

Schmid treated this species as being isolated [5]. Information about its distribution in China was provided by Malicky. So far, we do not have any specimens from the Provinces of Qinghai and Sichuan; therefore, we could not provide a *Diagnosis* nor any illustrations of the male genitalia of this species.

*Type country*: Russia.

*Distribution*: China (Qinghai, Sichuan); Russia.

#### 3.5.7. *Limnephilus* Species *incertae sedis*

Descriptions of the two Chinese species were based on females only: *L. incertus* Martynov, 1909, and *L. mclachlani* Martynov, 1909. Schmid treated them as species *incertae sedis* species [5]. We did not have the appropriate materials to revise them. Their systematic status remains unresolved.

*Limnephilus incertus* Martynov, 1909

*Limnophilus incertus* Martynov, 1909, 271–273, pl. 5, Figure 13, Figure 14 and Figure 15, female, China.

*Type country*: China.

*Distribution*: China (Qinghai).

*Limnephilus mclachlani* Martynov, 1909

*Limnophilus signifer* Martynov, 1909, 273–275, pl. 5, Figure 16, Figure 17 and Figure 18, female, China; preoccupied in *Limnephilus* by *Phryganea signifer* Zetterstedt, 1840

*Limnephilus mclachlani* Grigorenko, 2002, 114, replacement name.

*Type country*: China.

*Distribution*: China (Qinghai).

### 3.6. Distribution

A total of 17 *Limnephilus* species were recognized on the Chinese mainland, their distribution pattern is shown in Figure 19.

## 4. Discussion

*Limnephilus* larvae live predominantly in lentic habitats, but a few [21,31] have been collected in streams and cold springs, including the new species *Limnephilus deqianensis* **n. sp.**, which was collected in a cold stream during this study. Unfortunately, the investigation was performed very late, and all the larvae of the new species had pupated, so the only two larvae studied in this research were prepupae, acquired from the pupal cases. For this reason, we had no opportunity to observe the larval biology of the new species, including its feeding style, stadia, case-making behavior, etc., or its phenology and adult biology, such as the number of generations per year, the development speed, the synchronization of emergence, the life span of the adults, or the dispersal ability of the females. Therefore, further investigations are needed to determine these biological characteristics. We believe that the discovery of these biological features will enrich the stream macroinvertebrate traits datasets of China, which will allow them to be used in monitoring water quality and may even be helpful for inferring phylogenetical relationships among species in the genus *Limnephilus* and its higher taxa.

On the other hand, the taxonomic studies of the genus *Limnephilus* are mainly based on the structures of the male genitalia; species descriptions based only on female genitalia were rare, and certainly, there are some species known from both the male and female genitalia. For the species diagnoses based on the female genitalia, the abdominal segments and the vulvar scales were the most important diagnostic characteristics. However, the spermatheca is a chitinous organ inside the abdomen, potentially indicating a specificity among the species, as described by Nielsen [33]. Chuluunbat [47] applied his terminology for diagnosing the females of some Apataniidae species (also in the superfamily Limnephiloidea). Combined with the external morphology of the abdominal segments, the descriptions of the vulvar scales and the internal spermatheca can greatly improve our species identification ability and enrich the knowledge of our trichopteran fauna.

Terminology for the female genitalia presented by Schmid [27] has been widely used; however, he did not provide terms for the structure of the internal spermatheca. Nielsen [33] studied the female genital chamber of 26 species in detail and formulated terms for the structure of the internal spermatheca. In this article, we combined the terms of Schmid [32] and Nielson [33].

Finally, the morphological study of the larva and pupa of this species will be helpful in diagnosing *Limnephilus* species larvae and pupae in China, a common benthic macroinvertebrates group, which is useful for the biomonitoring of water quality. This morphological study of the male, female, larva, and pupa of *L*. *deqianensis* **n. sp.** will also provide morphological data for building hypotheses of phylogenetic relationships among species of this genus and its sister taxa. Perhaps because of the spumaline that covers the eggs of most Trichoptera, very little is known about the morphology of caddisfly eggs. Therefore, it is not likely that information about caddisfly embryos and other chorionic details will be useful for *Diagnosis* and phylogenetic inference in the near future.

## 5. Conclusions

In this study, most of the life stages of a new *Limnephilus* species, determined by *COI* sequencing, are described and illustrated. Ecological information regarding this new species is also provided.

Additionally, based on the male genitalia and following Schmid’s system, we revised all of the Chinese mainland *Limnephilus* species at the species group level while emphasizing their morphological characteristics. However, phylogenetic homologues or synapomorphies were not determined and cannot support the grouping of these species. Further phylogenetic studies between the species groups of *Limnephilus* are needed.

## Figures and Tables

**Figure 1 insects-13-00653-f001:**
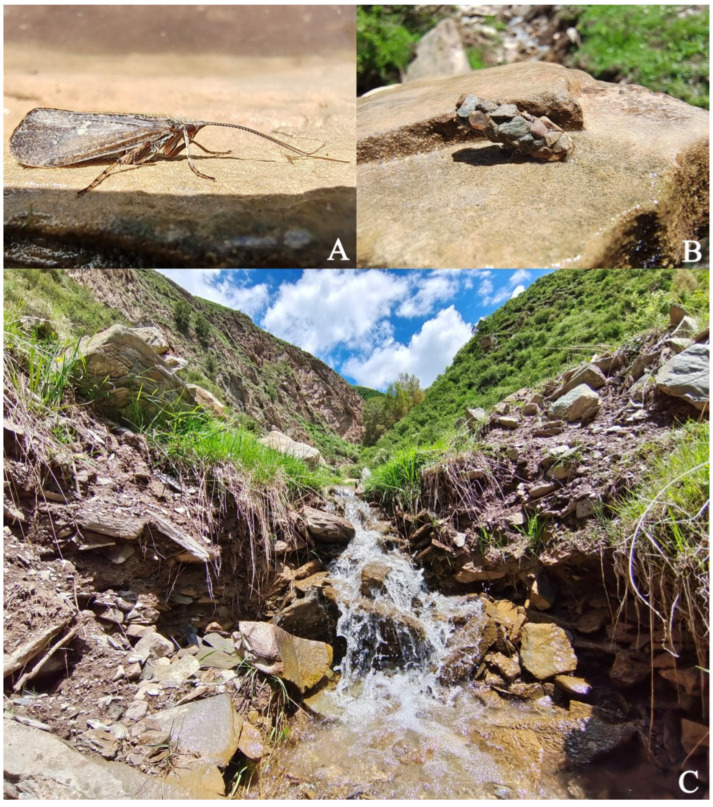
Ecological photos. (**A**) Male adult; (**B**) pupal case; and (**C**) habitat.

**Figure 2 insects-13-00653-f002:**
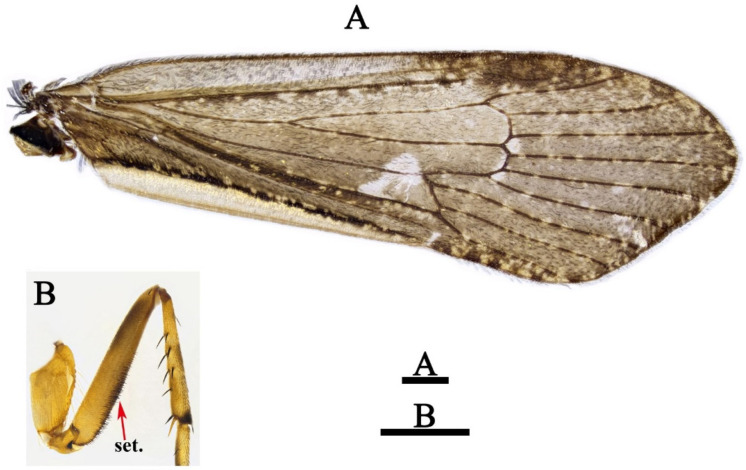
*Limnephilus deqianensis* **n. sp.**, male. (**A**) Forewing; (**B**) forefemur. Abbreviations: set. = setal brush. Scale bars: 1 mm.

**Figure 3 insects-13-00653-f003:**
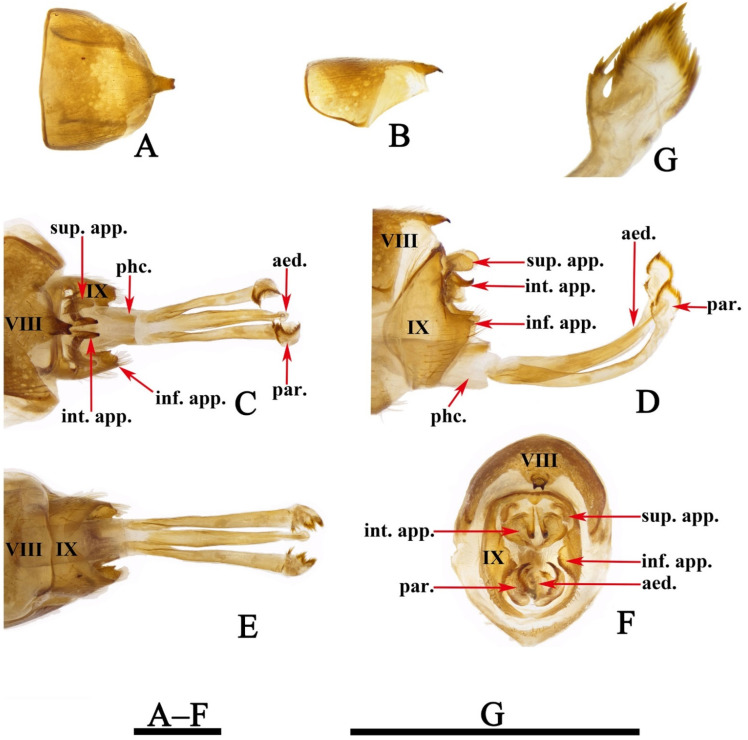
*Limnephilus deqianensis***n. sp.**, male genitalia. (**A**) Segment VIII, dorsal; (**B**) segment VIII, left lateral; (**C**) dorsal; (**D**) left lateral; (**E**) ventral; (**F**) caudal; (**G**) left lateral paramere subapex and small lobe, left lateral. Abbreviations: sup. app. = superior appendages (paired); int. app. = intermediate appendages (paired); inf. app. = inferior appendages (paired); phc. = phallocrypt; aed. = aedeagus; par. = paramere (paired). Scale bars: 1 mm.

**Figure 4 insects-13-00653-f004:**
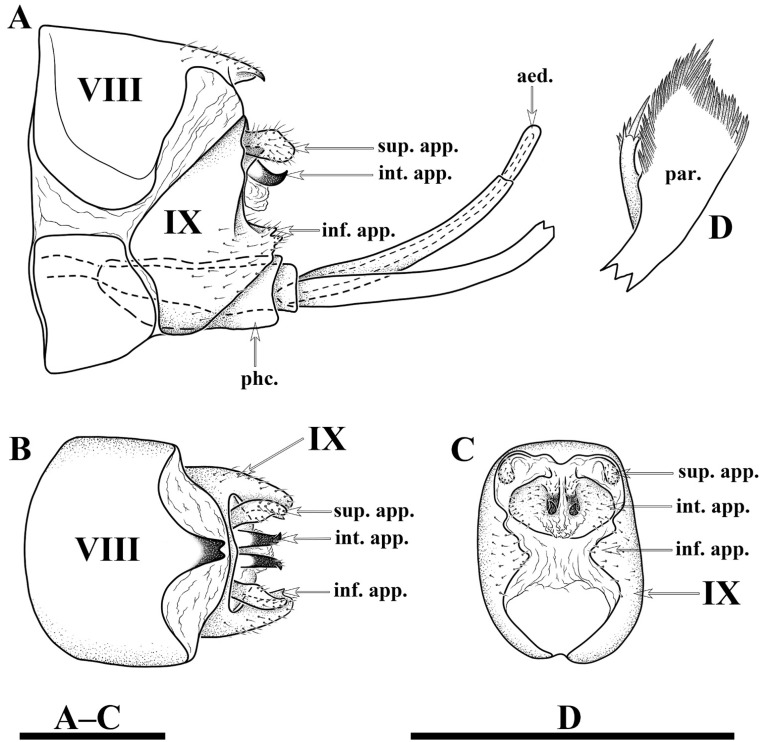
*Limnephilus deqianensis***n. sp.**, line drawing of male genitalia. (**A**) Left lateral; (**B**) dorsal; (**C**) caudal; (**D**) left lateral paramere subapex and small lobe, left lateral. Abbreviations: sup. app. = superior appendages (paired); int. app. = intermediate appendages (paired); inf. app. = inferior appendages (paired); phc. = phallocrypt; aed. = aedeagus; par. = paramere (paired). Scale bars: 1 mm.

**Figure 5 insects-13-00653-f005:**
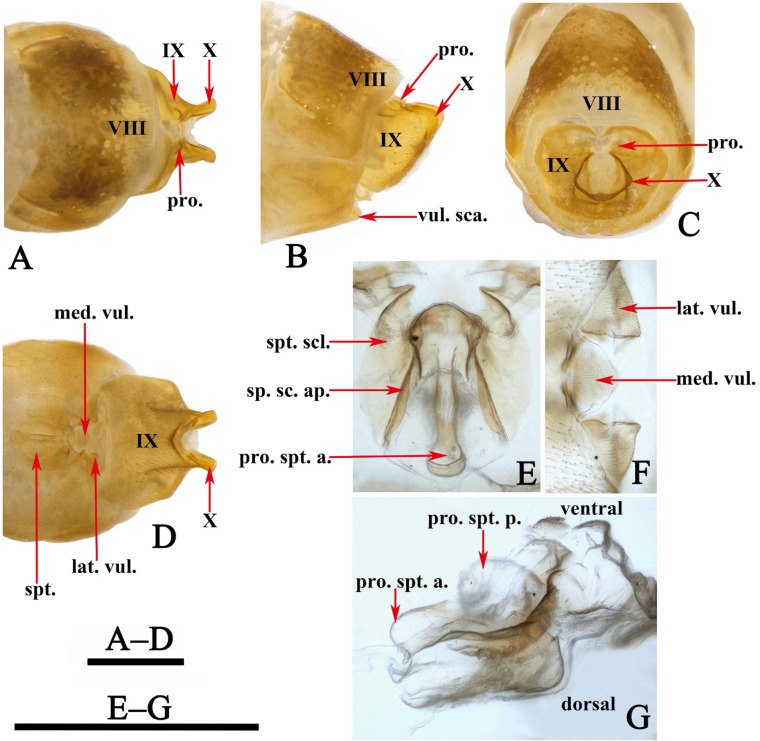
*Limnephilus deqianensis***n. sp.**, female genitalia. (**A**) Dorsal; (**B**) left lateral; (**C**) caudal; (**D**) ventral; (**E**) spermatheca, dorsal; (**F**) vulvar scales, ventral; (**G**) spermatheca, right lateral. Abbreviations: lat. vul. = lateral vulvar lobe (paired); med. vul. = median vulvar lobe; pro. = processus; pro. spt. = processus spermatheca; pro. spt. a. = anterior section of processus spermatheca; pro. spt. p. = posterior section of processus spermatheca; sp. sc. ap. = spermathecal sclerite apodeme. Scale bars: 1 mm.

**Figure 6 insects-13-00653-f006:**
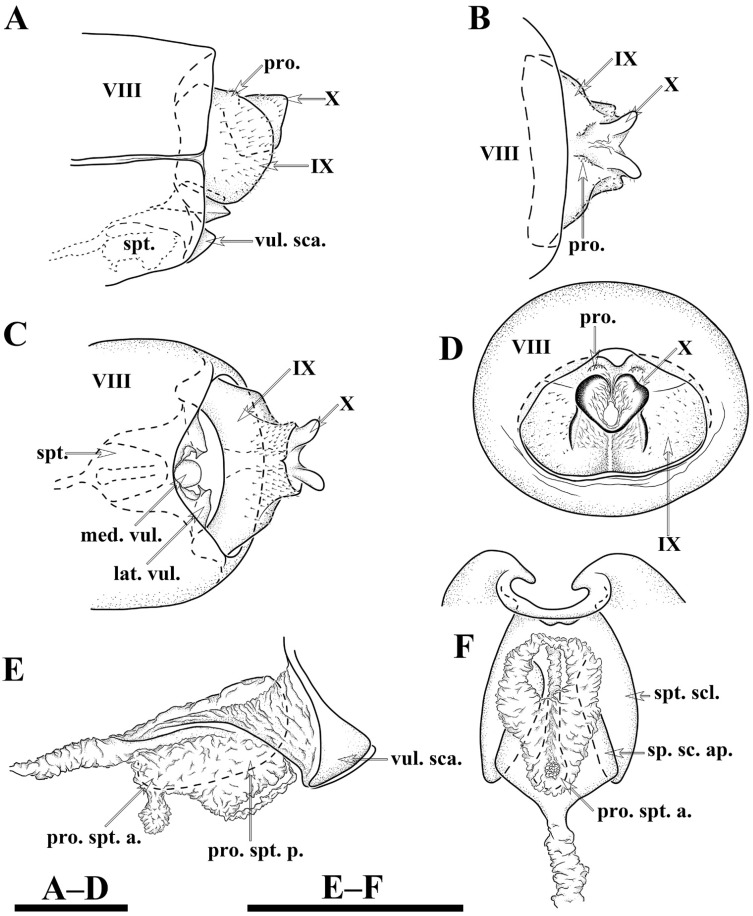
*Limnephilus deqianensis***n. sp.**, line drawing of female genitalia. (**A**) Left lateral; (**B**) dorsal; (**C**) ventral; (**D**) caudal; (**E**) spermatheca, left lateral; (**F**) spermatheca, ventral. Abbreviations: lat. vul. = lateral vulvar lobe (paired); med. vul. = median vulvar lobe; pro. = processus; pro. spt. = processus spermatheca; pro. spt. a. = anterior section of processus spermatheca; pro. spt. p. = posterior section of processus spermatheca; sp. sc. ap. = spermathecal sclerite apodeme. Scale bars: 1 mm.

**Figure 7 insects-13-00653-f007:**
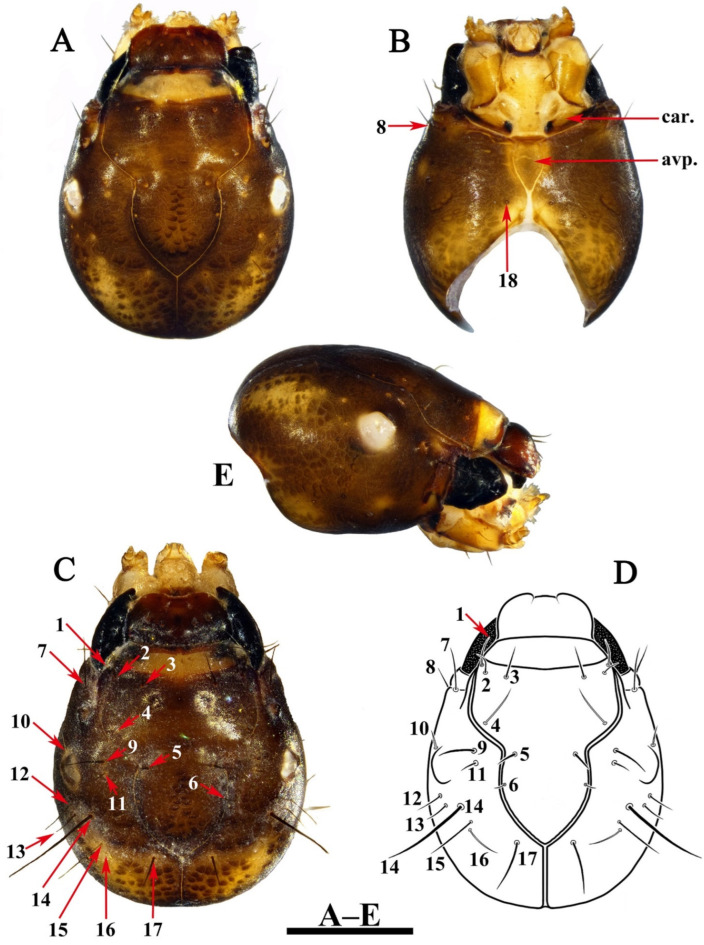
*Limnephilus deqianensis***n. sp.**, larva head. (**A**) Dorsal; (**B**) ventral; (**C**,**D**) primary setae on dorsal surface; (**E**) right lateral. Abbreviations: avp. = anterior ventral apotome; car = cardo. Scale bar: 1 mm.

**Figure 8 insects-13-00653-f008:**
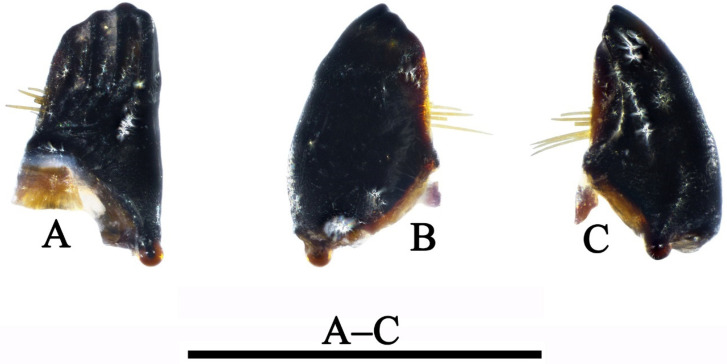
*Limnephilus deqianensis***n. sp.**, larva left mandible. (**A**) Mesal; (**B**), dorsal; (**C**) ventral. Scale bar: 1 mm.

**Figure 9 insects-13-00653-f009:**
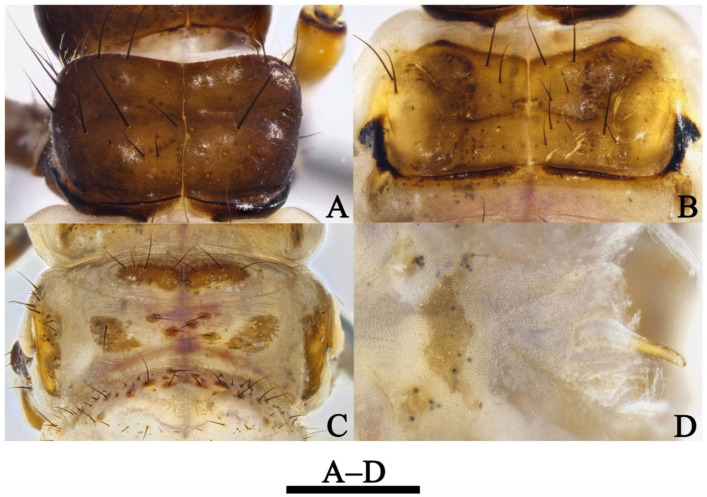
*Limnephilus deqianensis***n. sp.**, larval thorax. (**A**) Pronotum; (**B**) mesonotum; (**C**) metanotum; (**D**) prosternites and prosternal horn. Scale bar: 1 mm.

**Figure 10 insects-13-00653-f010:**
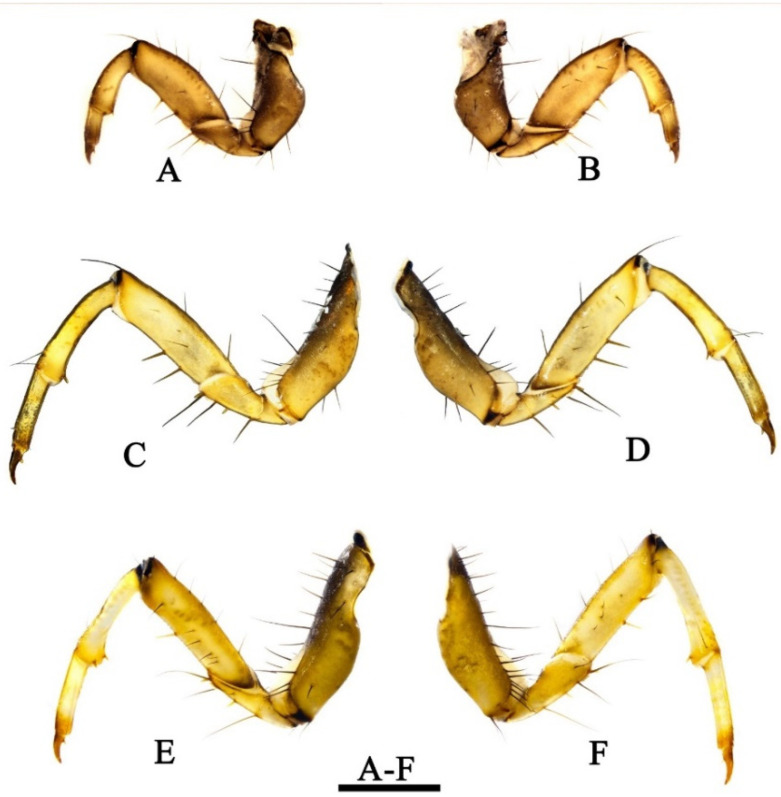
*Limnephilus deqianensis* **n. sp.**, larva legs. (**A**) Left foreleg posterior side; (**B**) left foreleg anterior side; (**C**) left midleg posterior side; (**D**) left midleg anterior side; (**E**) right hind leg anterior side; (**F**) right hind leg posterior side. Scale bar: 1 mm.

**Figure 11 insects-13-00653-f011:**
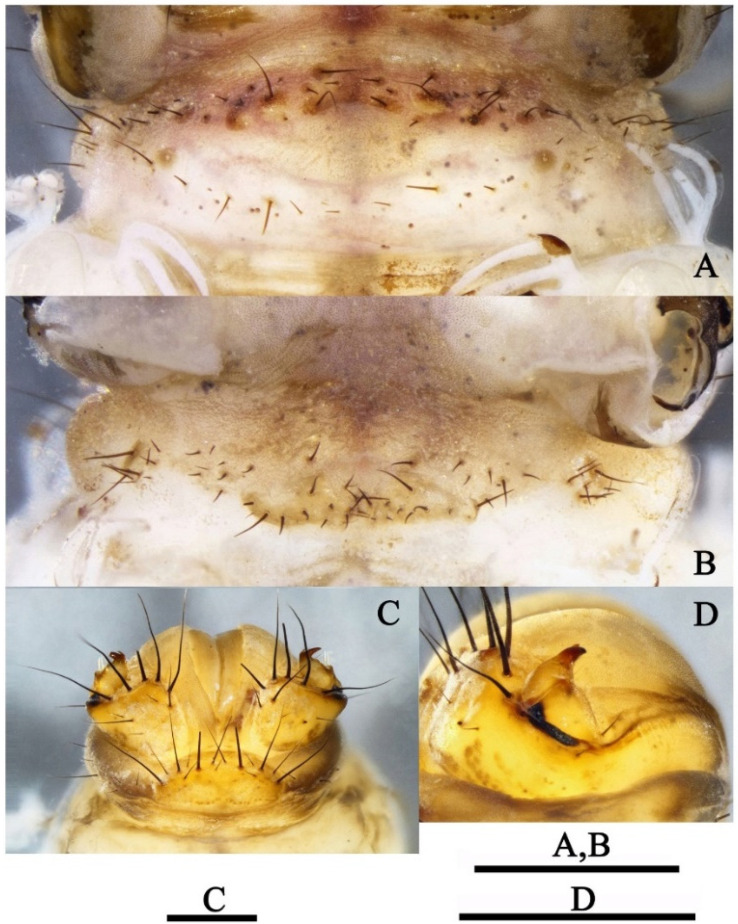
*Limnephilus deqianensis* **n. sp.**, larva abdomen. (**A**) Tergum of segment I; (**B**)sternum of segment I; (**C**) abdomoinal segment IX and anal prolegs; (**D**) left anal proleg (paired). Scale bars: 1 mm.

**Figure 12 insects-13-00653-f012:**
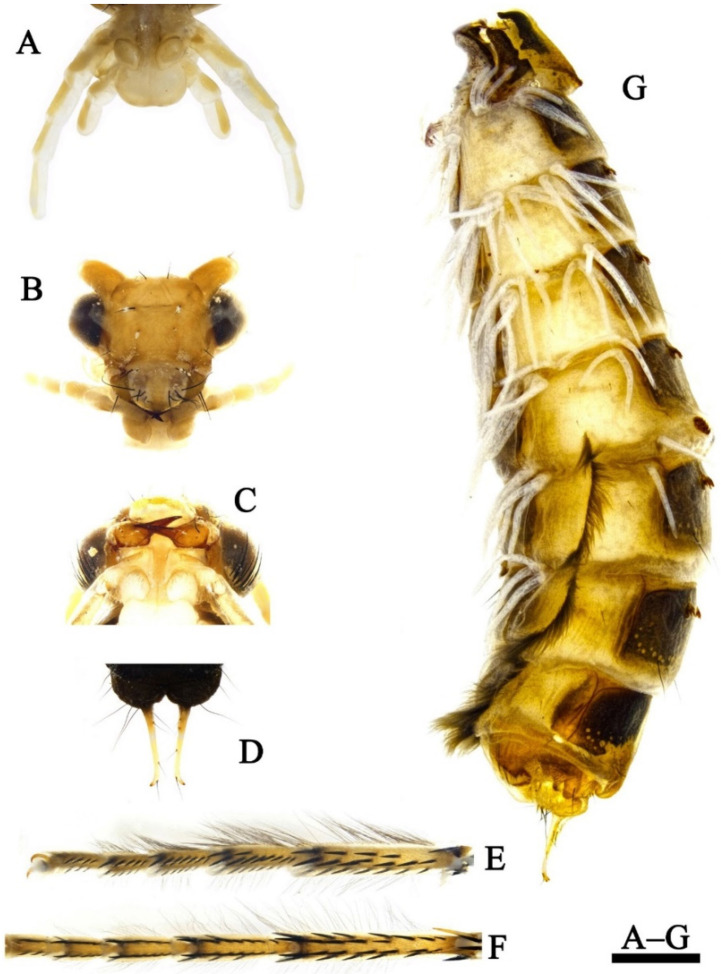
*Limnephilus deqianensis* **n. sp.**, pupa. (**A**) Maxillary and labial palps; (**B**) labrum; (**C**) mandibles; (**D**) anal appendages; (**E**) abdomen; (**F**) midtarsus; (**G**) hind tarsus. Scale bar: 1 mm.

**Figure 13 insects-13-00653-f013:**
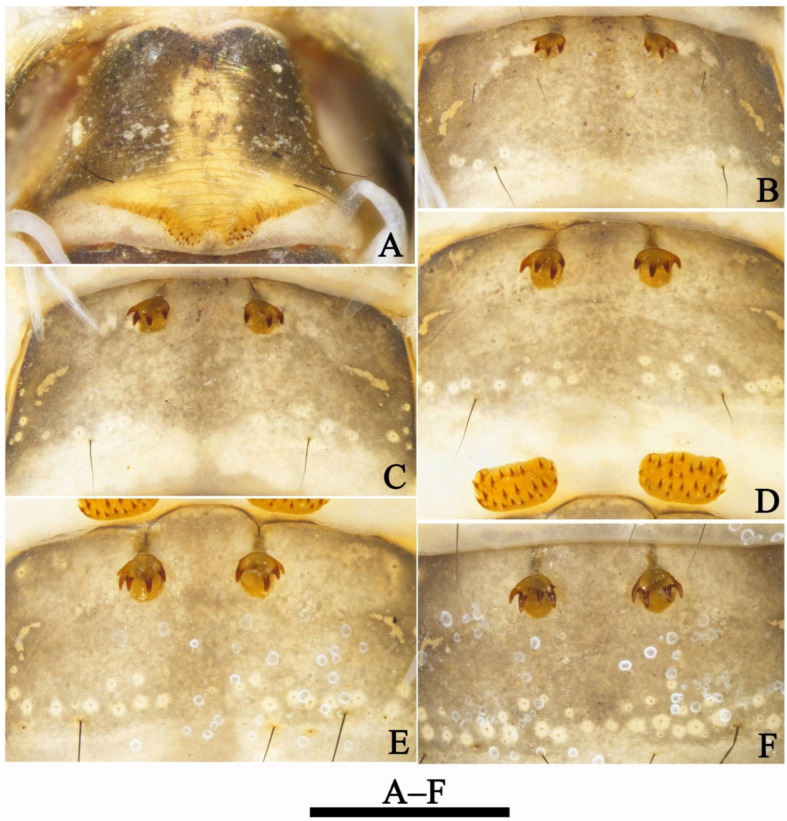
*Limnephilus deqianensis* **n. sp.**, pupa hook plates. (**A**) Ip; (**B**) IIIa; (**C**) IVa; (**D**) Va and Vp; (**E**) VIa; (**F**) VIIa. IIIa–VIIa = anterior hook plates of terga III–VII; Ip and Vp = posterior hook plates of terga I and V. Scale bar: 1 mm.

**Figure 14 insects-13-00653-f014:**
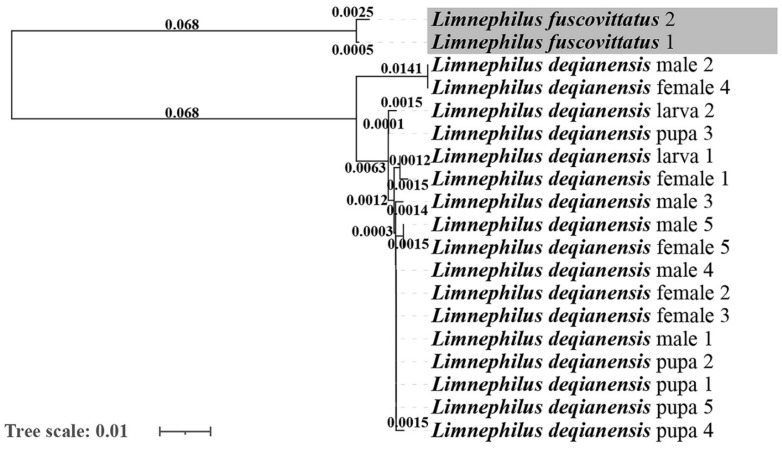
Neighbor-joining diagram based on sequenced data of the mitochondrial *COI* used to determine larva/female/male/pupa associations of *Limnephilus deqianensis* **n. sp.** Tree scale measures genetic distance.

**Figure 15 insects-13-00653-f015:**
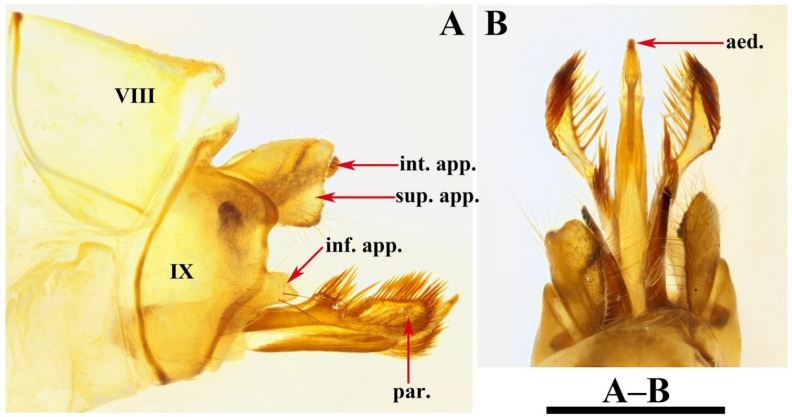
*Limnephilus correptus*, male genitalia. (**A**) Left lateral; (**B**) dorsal. Abbreviations: sup. app. = superior appendages (paired); int. app. = intermediate appendages (paired); inf. app. = inferior appendages (paired); aed. = aedeagus; par. = paramere. Scale bar: 1 mm.

**Figure 16 insects-13-00653-f016:**
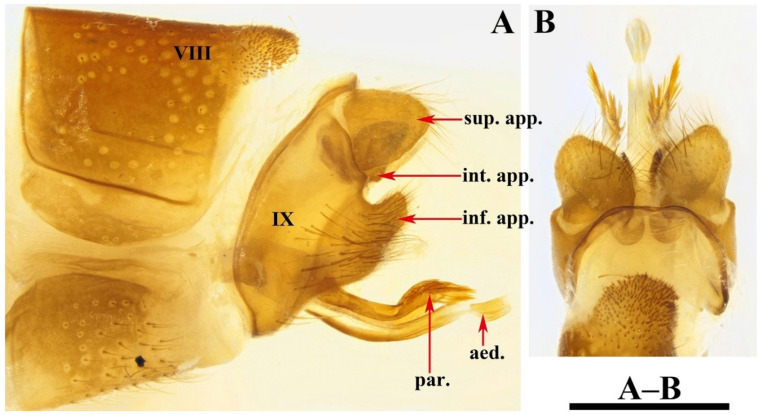
*Limnephilus fuscovittatus*, male genitalia. (**A**) Left lateral; (**B**) dorsal. Abbreviations: sup. app. = superior appendages (paired); int. app. = intermediate appendages (paired); inf. app. = inferior appendages (paired); aed. = aedeagus; par. = paramere. Scale bar: 1 mm.

**Figure 17 insects-13-00653-f017:**
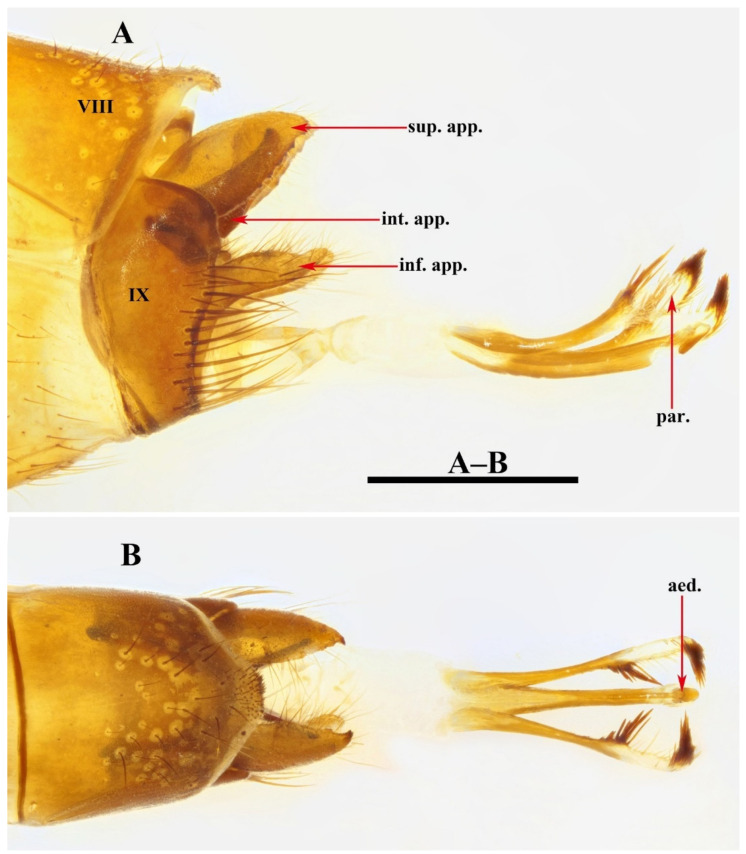
*Limnephilus zhejiangensis*, male genitalia. (**A**) Left lateral; (**B**) dorsal. Abbreviations: sup. app. = superior appendages (paired); int. app. = intermediate appendages (paired); inf. app. = inferior appendages (paired); aed. = aedeagus; par. = paramere. Scale bar: 1 mm.

**Figure 18 insects-13-00653-f018:**
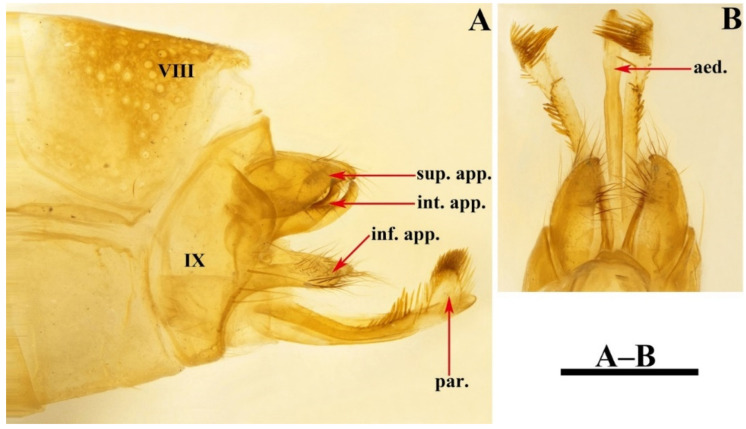
*Limnephilus distinctus*, male genitalia. (**A**) Left lateral; (**B**) dorsal. Abbreviations: sup. app. = superior appendages (paired); int. app. = intermediate appendages (paired); inf. app. = inferior appendages (paired); aed. = aedeagus; par. = paramere. Scale bar: 1 mm.

**Figure 19 insects-13-00653-f019:**
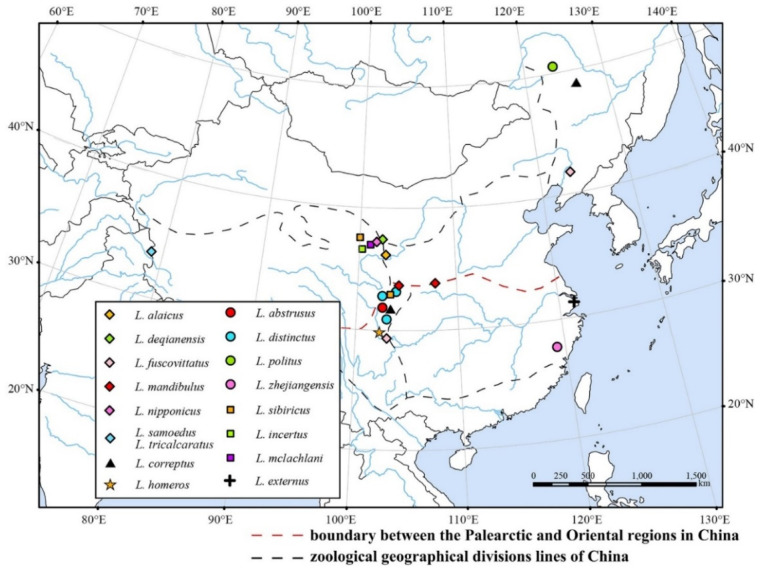
Distribution of *Limnephilus* species on the Chinese mainland.

**Table 1 insects-13-00653-t001:** Species of *Limnephilus* recorded from the Chinese mainland.

Species	Distribution
*correptus* McLachlan, 1880	China (Sichuan, Heilongjiang); Japan; Mongolia; North Korea; Russia
*distinctus* Tian & Yang, 1992	China (Sichuan)
*politus* McLachlan, 1865	China (Inner Mongolia); Britain; Czech Republic; Finland; Germany; Hungary; Kazakhstan; Netherlands; Norway; Poland; Russia; Sweden; Ukraine
*zhejiangensis* Leng & Yang, 2004	China (Zhejiang)
*abstrusus* McLachlan, 1872	China (Sichuan); Mongolia; Russia
*homeros* Malicky, 2011	China (Sichuan)
*externus* (Hagen, 1861)	China (Shanghai); America; Canada; Finland; Norway; Russia; Sweden
*fuscovittatus* Matsumura, 1904	China (Liaoning, Sichuan); Japan; Mongolia; North Korea; Russia.
*mandibulus* Yang & Yang, 2005	China (Shaanxi, Gansu)
*nipponicus* Schmid, 1964	China (Qinghai); Japan; Russia
*samoedus* (McLachlan, 1880)	China (Tibet); America; Mongolia; Russia
*tricalcaratus* (Mosely, 1936)	China (Tibet)
*alaicus* (Martynov, 1915)	China (Qinghai); Kazakhstan; Mongolia; Russia; Turkey
*mclachlani* Martynov, 1909	China (Qinghai)
*incertus* Martynov, 1909	China (Qinghai)
*sibiricus* Martynov, 1929	China (Qinghai, Sichuan); Russia

**Table 2 insects-13-00653-t002:** PCR primers used to sequence mt*COI* genes of *Limnephilus deqianensis* **n. sp.** in this study.

Primer	Sequence	Reference
LCO1490	GGTCAACAAATCATAAAGATATTGG	Folmer et al., 1994 [37]
HCO2198	TAAACTTCAGGGTGACAAAAAATCA	Folmer et al., 1994 [37]

**Table 3 insects-13-00653-t003:** Specimens used in larva-male-female-pupa associations of *Limnephilus deqianensis* **n. sp.**

Sample ID	Species	GenBank ID	Life Stage	Collection Site	Collection Date
QH21M1	*Limnephilus deqianensis*	ON834683	Adult (male)	De-qian town, Men-yuan county, Qinghai	2 July 2021
QH21M2	*Limnephilus deqianensis*	ON834679	Adult (male)	De-qian town, Men-yuan county, Qinghai	2 July 2021
QH21M3	*Limnephilus deqianensis*	ON834680	Adult (male)	De-qian town, Men-yuan county, Qinghai	2 July 2021
QH21M4	*Limnephilus deqianensis*	ON834682	Adult (male)	De-qian town, Men-yuan county, Qinghai	2 July 2021
QH21M5	*Limnephilus deqianensis*	ON834681	Adult (male)	De-qian town, Men-yuan county, Qinghai	2 July 2021
QH21F1	*Limnephilus deqianensis*	ON834677	Adult (female)	De-qian town, Men-yuan county, Qinghai	2 July 2021
QH21F2	*Limnephilus deqianensis*	ON834673	Adult (female)	De-qian town, Men-yuan county, Qinghai	2 July 2021
QH21F3	*Limnephilus deqianensis*	ON834674	Adult (female)	De-qian town, Men-yuan county, Qinghai	2 July 2021
QH21F4	*Limnephilus deqianensis*	ON834675	Adult (female)	De-qian town, Men-yuan county, Qinghai	2 July 2021
QH21F5	*Limnephilus deqianensis*	ON834676	Adult (female)	De-qian town, Men-yuan county, Qinghai	2 July 2021
QH21P1	*Limnephilus deqianensis*	ON834687	Pupa	De-qian town, Men-yuan county, Qinghai	2 July 2021
QH21P2	*Limnephilus deqianensis*	ON834684	Pupa	De-qian town, Men-yuan county, Qinghai	2 July 2021
QH21P3	*Limnephilus deqianensis*	ON834685	Pupa	De-qian town, Men-yuan county, Qinghai	2 July 2021
QH21P4	*Limnephilus deqianensis*	ON834686	Pupa	De-qian town, Men-yuan county, Qinghai	2 July 2021
QH21P5	*Limnephilus deqianensis*	ON834689	Pupa	De-qian town, Men-yuan county, Qinghai	2 July 2021
QH21L1	*Limnephilus deqianensis*	ON834678	Larva	De-qian town, Men-yuan county, Qinghai	2 July 2021
QH21L2	*Limnephilus deqianensis*	ON834688	Larva	De-qian town, Men-yuan county, Qinghai	2 July 2021
ID06902	*Limnephilus fuscovittatus*	KX104327	n/a	Mongolia: Arhangay, Tariat, Urd Terkhiin Gol	25 July 2004
ID06904	*Limnephilus fuscovittatus*	KX103754	n/a	Mongolia: Hovsgol, Tsagaan-Uur, Uur Gol	16 July 2005

**Table 4 insects-13-00653-t004:** Number of types of tracheal gills on abdominal segments II–VII of final instar larvae of *Limnephilus deqianensis* **n. sp.** Positions abbreviated as: A = anterior, D = dorsal, L = lateral, P = posterior, V = ventral.

Gill	II	III	IV	V	VI	VII
AD	3	3	3	2	1	0
PD	3	3	2	1	0	0
ADL	3	3	1	3	0	0
AVL	3	2	1	0	0	0
AV	3	3	3	2	2	2
PV	3	3	3	2	2	2

**Table 5 insects-13-00653-t005:** Number of types of tracheal gills on abdominal segments II–VII of pupae of *Limnephilus deqianensis* **n. sp.** Positions abbreviated as: A = anterior, D = dorsal, L = lateral, P = posterior, V = ventral.

Gill	II	III	IV	V	VI	VII
ADL	3	3	3	2	0	0
PDL	3	3	2	0	0	0
AL	3	3	2	1	0	0
PL	3	2	1	0	0	0
AVL	3	3	3	2	2	1
PVL	3	3	3	2	2	0

**Table 6 insects-13-00653-t006:** Morphological characteristics of superior and inferior appendages of the eight species in the *L*. *perpusillus* Species Group *nomen novum.* Abbreviations: sup. app. = superior appendages; inf. app. = inferior appendages.

Species	*perpusillus*	*janus*	*hyalinus*	*major*	*secludens*	*ademus*	*acnestus*	*homeros*
Incision of sup. app.	without	without	without	with	with	with	with	with
Fused portion of inf. app.	almost invisible	almost invisible	narrow	narrow	narrow	narrow	narrow	narrow

## Data Availability

The data, holotype specimen, and other voucher specimens from this research were deposited in the Insect Classification and Aquatic Insect Laboratory, College of Plant Protection, Nanjing Agricultural University, Nanjing, China and Department of Entomology and Nematology, University of California Davis, CA, USA.

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
