# Peer review of "A New Species of *Limnephilus* (Insecta: Trichoptera: Limnephilidae) from China, with Revision of the Genus *Limnephilus* on the Chinese Mainland [Author-notes fn1-insects-13-00653]"

_insects, 2022, doi:10.3390/insects13070653_

Round 1

Reviewer 1 Report

I congratulate the authors on their efforts to describe all life forms (other than eggs) of this new species. Editorial recommendations are marked in the attached version of the manuscript.

Because of their complexity, the photographs particularly of male and female genitalia are not adequate for illustrating details of several diagnostic characters of this species. Some photographs of larvae are also a problem because some diagnostic setae are not visible in them. I recommend that labeled line drawings should accompany the photographs and descriptions of at least male and female genitalia in order to help assure understanding of the diagnostic characters.

Tables 3 and 4 are not in standard format and lack important details. Please see the comments in the attached reviewed manuscript for an explanation of the format that should be used.

Author Response

Dear Reviewer,

Thank you so much for your valuable comments and suggestions on our manuscript. We have modified and improved our manuscript according to your kind advice and detailed suggestions. Overall, we have made extensive editing of the English language, provided line drawings of male genitalia, female genitalia, and primary setae on the larval head, and provided a standard representation of gills. In addition, we have summarized the species of Limnephilus found within the Chinese Mainland. Every one of your suggestions has been addressed. Enclosed please find out the Responses to Reviewer 1 Comments.

Special thanks to you for your careful edit of the language. Thank you for the time and effort that you have put into reviewing our manuscript.

Looking forward to hearing from you soon.

Best regards

Sincerely yours

Haoming Zang, Beixin Wang, and Changhai Sun on behalf of all co-authors

Response to Reviewer 1 Comments

Point 1: The terms “new species” and “COI” are included in the title of the article. They will be found in all digital searches so that they are redundant here and should be replaced.

Response 1: We have changed the title and removed the ‘’new species’’ from the keywords. Thank you for pointing out this deficiency.

Point 2: As explained in comments below, because of their complexity, the photographs particularly of male and female genitalia are not adequate for illustrating details of several diagnostic characters of this species. Some photographs of larvae are also a problem because some diagnostic setae are not visible in them. I recommend that labeled line drawings should accompany the photographs and descriptions of at least male and female genitalia in order to help assure understanding of the diagnostic characters.

Response 2: We have added line drawings of male genitalia, female genitalia, and primary setae on the larval head dorsal surface. Thank you for pointing out this deficiency.

Point 3: The editor should assure that this small table appears together on one page.

Response 3: This small table should appear on a new page. Thank you for pointing out this deficiency.

Point 4: The body lengths, antennal lengths, and wing lengths of males and females are usually different. Please provide these data separately for males and females.

Response 4: Yes, you are right. We have added the length range of these structures and these data were provided separately for males and females. Thank you for pointing out this deficiency.

Point 5: Please provide the range of body lengths measured for these 20 specimens, with body lengths for males and females provided separately.

Response 5: We have provided the range of body lengths for males and females separately. Thank you for pointing out this deficiency.

Point 6: Detailed dorsal and lateral illustrations are needed to show the details of the intermediate appendages. The details mentioned here are not visible in Figs 3C, 3D, and 3 F.

Response 6: We have provided detailed line drawings of intermediate appendages. Thank you for pointing out this deficiency.

Point 7: Male genitalia of most Trichoptera species, including this species, are too complicated in their 3-dimensional structure to be represented by photographs alone. Interpretive line drawings are needed to assure adequate understanding of these complex genitalia and their parts. Please provide line drawings to accompany these photographs to show details of all structures mentioned in the description.

Response 7: We have provided line drawings for male and female genitalia. Thank you for pointing out this deficiency.

Point 8: Except for those having white background, the margins of external structures of female genitalia in Figures 4A–4D are very obscure. These figures should be accompanied or replaced by line drawings.

Response 8: We have provided line drawings for female internal spermatheca. Thank you for pointing out this deficiency.

Point 9: These abbreviations should be arranged in alphabetical order.

Response 9: We have rearranged the order of these abbreviations. Thank you for pointing out this deficiency.

Point 10: Please explain in Material and Methods how you detrmined that the larva you are describing is the final instar (e.g., by comparison with metamorphotype larval sclerites remaining in the pupal cocoons from your reared pupae).

Response 10: We have explained our determination in Material and Methods. Thank you for pointing out this deficiency.

Point 11: Is this correct? In life, the eye is usually black with a white border. I suspect that you are seeing only white eyes because these specimens are prepupae which have already begun to molt to pupae.

Response 11: Yes, you are right. We have explained the possible cause of the white eyes. Thank you for pointing out this deficiency.

Point 12: In English, “thin” can refer to an object with one small dimension (e.g., a sheet of paper). However, “fine” refers unequivocably to an object with two small dimensions (e.g., a hair) among the three possible dimensions.

Response 12: We have changed the word in our manuscript. Thank you very much for explaining in detail.

Point 13: There is an error here. “. . . setae 10 finer and shorter than 10 . . .”

Response 13: We have corrected this mistake. Thank you for pointing out this error.

Point 14: Primary setae 18 are true setae and not pores. Apparently, these setae have been broken away so that only their alveoli are present.

Response 14: Thank you very much for explaining in detail. We have corrected this mistake in our manuscript.

Point 15: Label the cardo in Fig. 5B.–

Response 15: We have labeled the cardo. Thank you for pointing out this deficiency.

Point 16: These setae are not visible in Figure 7A. Are they occurring only on the posterior margin? If they are not restricted to the posterior margin, their description should not follow immediately after the description of the posterior margin and, instead, should follow immediately after the description of the several black, spike-like setae.

Response 16: This is an error. We have deleted this sentence in our manuscript.

Point 17: Characters should be discussed in spatial order from anterior to posterior. In this situation, these characters should not follow the description of the posterior margin.

Response 17: Yes, you are right. We have deleted this sentence here.

Point 18: Only 2 setae are visible on apical segment in Fig. 8A, suggesting that an improved photograph and/or a line drawing are needed.

Response 18: Yes, you are right. We have replaced these figures with improved photograph. Thank you for pointing out this deficiency.

Point 19: Only two setae visible on ventral margin in Figs 8A, 8B. An improved photograph and/or a line drawing are needed.

Response 19: Yes, you are right. We have replaced these figures with improved photographs. Thank you for pointing out this deficiency.

Point 20: This table is difficult to understand. For example, are there really 6 trifid gills on abdominal segment II? If so, what is their distribution among the six usual locations: anteriorly and posteriorly in dorsolateral, lateral, and ventrolateral locations?

Response 20: Yes, you are right. We have provided a standard representation of gills. Thank you for pointing out this deficiency.

Point 21: A much-more standard representation of gills would be a gill map representing one side of the abdomen, with segment numbers and anterior or posterior positions in columns (IIa, IIp, IIIa, IIIp, etc.) and gill positions on those segments in three rows (dorso-lateral, lateral, and ventro-lateral). In each column, provide “0” or “1” or “2” or “3” in each of the three positions to indicate that the gill in that position is not present (“0”) or is present with 1 or 2 or 3 filaments.

Response 21: Yes, you are right. We have provided a standard representation of gills. Thank you for pointing out this deficiency. Thank you very much for explaining in detail.

Point 22: How many setae?

Response 22: We have provided the number of setae. Thank you for pointing out this deficiency.

Point 23: Again, this information is not standard and is difficult to understand. Please provide these details with the format explained in the comment for Table 3.

Response 23: We have provided this information in a standard format. Thank you for pointing out this deficiency.

Point 24: Please provide a reference in which the female of this species was described and illustrated.

Response 24: We have provided the reference. Thank you for pointing out this deficiency.

Point 25: What appendage does L. deqianensis lack but that is present in L. alienus?

Response 25: We have replaced L. aliens with Limnephilus primoryensis Nimmo, 1995 because the female genitalia of L. primoryensis is closer to the new species.

Reviewer 2 Report

This paper represents a very thorough and detailed description of a new species and a very nice summary of the species of the genus found within China. It would benefit from some light editing to the language, grammar, and punctuation. Concerns and suggestions have been outlined in the attached PDF.

Line 25: ‘Pupal case’, not ‘pupalcase’. 2 words, not 1. Needs to be fixed throughout the manuscript.

Line 35: ‘region’, not ‘Region’. Does not need to be capitalized, needs to be fixed throughout the manuscript.

Line 39: ‘monotypic’, not ‘monospecies-groups’

Lines 39-40: what is the difference here between monotypic, isolated, and incertae sedis? I’m not familiar with a species being ‘isolated’ in this context and I’m not sure what this term is intended to indicate in this manuscript.

Lines 44-50: this parenthetical statement is extremely long and confusing to read. It should be re-written as complete sentences, not as a parenthetical.

Lines 45-46: ‘species group’, not ‘Species Group’. Unnecessary capitalization, needs to be fixed throughout the manuscript.

Line 47: When did Schmid make this placement? This needs a citation.

Line 48: When did Nimmo perform this transfer? Needs a citation.

Line 49: When did Nimmo make this placement? This needs a citation.

Line 50: Need a citation for Ruiter’s action.

Line 70: The members of Limnephilus . . .

Line 73: The genus is particularly well . . .

Lines 81-84: messy sentence with awkward wording, needs to be cleaned up

Line 88: Additional specimens studied . . .

Line 92: . . . captured by hand during collection . . .

Table 1: Distribution spelled incorrectly in table header

Table 1: again, what is the difference between an ‘isolated species’ and one considered incertae sedis?

Table 1: I understand that this paper is focused on the species of Limnephilus that occur in China, but could all species regardless of their distribution be listed in the table? It is somewhat confusing in the later sections to read that there are X number of species in the species groups, but then have no mention made at all of the species not found in China.

Lines 128-129: COI sequences are uploaded to GenBank and accession numbers are included in Table 3.

Line 138: It would be extremely helpful to have figures to refer to when using this key. Is it possible to either refer to figures already included in the manuscript or to produce some sort of general diagrams of the morphology being described?

Line 158: It would be helpful to formally and explicitly acknowledge the new combination in the header

Lines 165 and onward: These numbered lists of taxa and morphological characters are hard to follow, could they be arranged into some sort of table? Here and elsewhere throughout the manuscript?

Lines 232-237: These sentences are messy and need to be cleaned up.

Line 261: You should explicitly refer to Figs. 3-12 here.

Line 272: What is 14.2 mm long? The antennae, the length of the body?

Figure 4 legend: Indicate the abdominal segments included in C-F. I assume these include abdominal segments IX and X? This needs to be addressed throughout the manuscript.

Figure 4: the scale bars are very awkward inserted into the middle of the figure, can they be moved to an edge or a side somewhere?

Figure 6: if there is only 1 scale bar needed in this figure, rather than the 2 bars needed in previous figures, does it need to be labeled A-D? This needs to be addressed throughout the manuscript.

Line 336/Figure 6: the primary setae are hard to see in the photo, even with the red arrows pointing them out. Could the photo resolution or contrast be adjusted to improve their visibility? Or could a general diagram of setal position be provided?

Figure 10: could the A-B scale bar be inserted into views A and B, the way the C bar is part of the C view? Could the D bar also be inserted into the D view?

Figure 11 legend: ‘palps’, not palp

Figure 11: awkward placement of the scale bar

Figure 13: the manuscript text indicates that the scale bar is measuring genetic distances, but this is not mentioned in the figure legend. Please explicitly state in the figure legend what is being measured by the scale bar.

Line 491: It feels very odd/abrupt to go back to the diagnoses for the species groups after the thorough new species description. Can this be rearranged so that the diagnoses of the species groups can flow uninterrupted?

Lines 503-507: messy sentences, need to be cleaned up.

Line 509: the latter what? What are you referring to as ‘latter’?

Lines 624-626: messy sentences, need to be cleaned up. They are not grammatically correct, complete sentences.

Figure 17: if the red dashed line on the map indicates the boundary between the Palearctic and Oriental regions, what do the dashed lines indicate?

Lines 644-667: Messy paragraphs, awkward to read. Needs to be cleaned up and re-written.

Lines 668-672: This information was already outlined in an earlier section, does it need to be mentioned again here?

Author Response

Dear Reviewer,

Thank you so much for your valuable comments and suggestions on our manuscript. We have modified and improved our manuscript according to your kind advice and detailed suggestions. Overall, we have made extensive editing of the English language, and every one of your suggestions has been addressed. Enclosed please find out the Responses to Reviewer 2 Comments.

Thank you for the time and effort that you have put into reviewing our manuscript.

Looking forward to hearing from you soon.

Best regards

Sincerely yours

Haoming Zang, Beixin Wang, and Changhai Sun on behalf of all co-authors

Response to Reviewer 2 Comments

Point 1: Line 25: ‘Pupal case’, not ‘pupalcase’. 2 words, not 1. Needs to be fixed throughout the manuscript.

Response 1: Yes, you are right. We have corrected this mistake. Thank you for pointing out this error.

Point 2: Line 35: ‘region’, not ‘Region’. Does not need to be capitalized, needs to be fixed throughout the manuscript.

Response 2: We have corrected this mistake throughout the manuscript. Thank you for pointing out this error.

Point 3: Line 39: ‘monotypic’, not ‘monospecies-groups’

Response 3: Yes, you are right. We have corrected this mistake. Thank you for pointing out this error.

Point 4: Lines 39-40: what is the difference here between monotypic, isolated, and incertae sedis? I’m not familiar with a species being ‘isolated’ in this context and I’m not sure what this term is intended to indicate in this manuscript.

Response 4: Isolated refers to the species with its genitalia highly specialized and have not been ascribed to species groups; incertae sedis refers to the species with its placement uncertain. We have explained these words in our manuscript. Thank you for pointing out this deficiency.

Point 5: Lines 44-50: this parenthetical statement is extremely long and confusing to read. It should be re-written as complete sentences, not as a parenthetical.

Response 5: We have revised this paragraph. Thank you for pointing out this deficiency.

Point 6: Lines 45-46: ‘species group’, not ‘Species Group’. Unnecessary capitalization, needs to be fixed throughout the manuscript.

Response 6: We have corrected this mistake throughout the manuscript. Thank you for pointing out this error.

Point 7: Line 47: When did Schmid make this placement? This needs a citation. Line 48: When did Nimmo perform this transfer? Needs a citation.

Response 7: We have added this citation. Thank you for pointing out this deficiency.

Point 8: Line 49: When did Nimmo make this placement? This needs a citation.

Response 8: We have added this citation. Thank you for pointing out this deficiency.

Point 9: Line 50: Need a citation for Ruiter’s action.

Response 9: We have added this citation. Thank you for pointing out this deficiency.

Point 10: Line 70: The members of Limnephilus . . .

Response 10: We have done the replacement. Thank you for pointing out this deficiency.

Point 11: Line 73: The genus is particularly well . . .

Response 11: We have done the replacement. Thank you for pointing out this deficiency.

Point 12: Lines 81-84: messy sentence with awkward wording, needs to be cleaned up

Response 12: We have modified this sentence. Thank you for pointing out this deficiency.

Point 13: Line 88: Additional specimens studied . . .

Response 13: We have done the replacement. Thank you for pointing out this deficiency.

Point 14: Line 92: . . . captured by hand during collection . . .

Response 14: Yes, you are right. We have corrected this mistake. Thank you for pointing out this error.

Point 15: Table 1: Distribution spelled incorrectly in table header

Response 15: Yes, you are right. We have corrected this mistake. Thank you for pointing out this error.

Point 16: Table 1: again, what is the difference between an ‘isolated species’ and one considered incertae sedis?

Response 16: Isolated refers to the species with its genitalia highly specialized and have not been ascribed to species groups; incertae sedis refers to the species with its placement uncertain. We have explained these words in our manuscript. Thank you for pointing out this deficiency.

Point 17: Table 1: I understand that this paper is focused on the species of Limnephilus that occur in China, but could all species regardless of their distribution be listed in the table? It is somewhat confusing in the later sections to read that there are X number of species in the species groups, but then have no mention made at all of the species not found in China.

Response 17: We have listed all species in each species group. Thank you for pointing out this deficiency.

Point 18: Lines 128-129: COI sequences are uploaded to GenBank and accession numbers are included in Table 3.

Response 18: We have done the replacement. Thank you for pointing out this deficiency.

Point 19: Line 138: It would be extremely helpful to have figures to refer to when using this key. Is it possible to either refer to figures already included in the manuscript or to produce some sort of general diagrams of the morphology being described?

Response 19: We have referred all species groups in this key to figures or references included in our manuscript. Some species groups are not photographed because of a lack of specimens. Fortunately, they have figures in their references. Thank you for pointing out this deficiency.

Point 20: Line 158: It would be helpful to formally and explicitly acknowledge the new combination in the header

Response 20: We have done the modification. Thank you for pointing out this deficiency.

Point 21: Lines 165 and onward: These numbered lists of taxa and morphological characters are hard to follow, could they be arranged into some sort of table? Here and elsewhere throughout the manuscript?

Response 21: We have added a table including these species and morphological. Thank you for pointing out this deficiency.

Point 22: Lines 232-237: These sentences are messy and need to be cleaned up.

Response 22: We have modified these sentences. Thank you for pointing out this deficiency.

Point 23: Line 261: You should explicitly refer to Figs. 3-12 here.

Response 23: Done. Thank you for pointing out this deficiency.

Point 24: Line 272: What is 14.2 mm long? The antennae, the length of the body?

Response 24: It is the length of the antennae.

Point 25: Figure 4 legend: Indicate the abdominal segments included in C-F. I assume these include abdominal segments IX and X? This needs to be addressed throughout the manuscript.

Response 25: We have indicated seg IX–X throughout our manuscript. Thank you for pointing out this deficiency.

Point 26: The scale bars are very awkward inserted into the middle of the figure, can they be moved to an edge or a side somewhere?

Response 26: We have moved the scale bars to bottom side of the figure. Thank you for pointing out this deficiency.

Point 27: Figure 6: if there is only 1 scale bar needed in this figure, rather than the 2 bars needed in previous figures, does it need to be labeled A-D? This needs to be addressed throughout the manuscript.

Response 27: Thank you for your suggestion. However, referring to an article published in this journal (https://doi.org/10.3390/insects12121089), we decided to keep it.

Point 28: Line 336/Figure 6: the primary setae are hard to see in the photo, even with the red arrows pointing them out. Could the photo resolution or contrast be adjusted to improve their visibility? Or could a general diagram of setal position be provided?

Response 28: We have provided a line drawing of primary setae on the larval head dorsal surface and improved the visibility of figures. Thank you for pointing out this deficiency.

Point 29: Figure 10: could the A-B scale bar be inserted into views A and B, the way the C bar is part of the C view? Could the D bar also be inserted into the D view?

Response 29: We have moved all scale bars to the bottom side of the figure. Thank you for pointing out this deficiency.

Point 30: Figure 11 legend: ‘palps’, not palp

Response 30: Yes, you are right. We have corrected this mistake. Thank you for pointing out this error.

Point 31: Figure 11: awkward placement of the scale bar

Response 31: We have modified this figure. Thank you for pointing out this error.

Point 32: Figure 13: the manuscript text indicates that the scale bar is measuring genetic distances, but this is not mentioned in the figure legend. Please explicitly state in the figure legend what is being measured by the scale bar.

Response 32: We have done the statement. Thank you for pointing out this deficiency.

Point 33: Line 491: It feels very odd/abrupt to go back to the diagnoses for the species groups after the thorough new species description. Can this be rearranged so that the diagnoses of the species groups can flow uninterrupted?

Response 33: We have rearranged the manuscript. Thank you for pointing out this deficiency.

Point 34: Lines 503-507: messy sentences, need to be cleaned up.

Response 34: We have modified these sentences. Thank you for pointing out this deficiency.

Point 35: Line 509: the latter what? What are you referring to as ‘latter’?

Response 35: We have replaced ‘latter’ with L. alaicus. Thank you for pointing out this deficiency.

Point 36: Lines 624-626: messy sentences, need to be cleaned up. They are not grammatically correct, complete sentences.

Response 36: We have modified these sentences. Thank you for pointing out this deficiency.

Point 37: Figure 17: if the red dashed line on the map indicates the boundary between the Palearctic and Oriental regions, what do the dashed lines indicate?

Response 37: We have added the indication of the dashed lines. Thank you for pointing out this deficiency.

Point 38: Q38 Lines 644-667: Messy paragraphs, awkward to read. Needs to be cleaned up and re-written.

Response 38: We have modified these paragraphs. Thank you for pointing out this deficiency.

Point 39: Lines 668-672: This information was already outlined in an earlier section, does it need to be mentioned again here?

Response 39: We have simplified this passage. Thank you for pointing out this deficiency.

Reviewer 3 Report

The paper is close to being in final form. There are just a number of minor issues scattered in the attached PDF that need to be corrected.

Author Response

Dear Reviewer,

Thank you so much for your valuable comments and suggestions on our manuscript. We have modified and improved our manuscript according to your kind advice and detailed suggestions. We have listed the changes to our manuscript below.

Thank you for the time and effort that you have put into reviewing our manuscript.

Looking forward to hearing from you soon.

Best regards

Sincerely yours

Haoming Zang, Beixin Wang, and Changhai Sun on behalf of all co-authors

  1. specimen

Response: We have corrected this mistake. Thank you for pointing out this error.

  1. The BX41 is a light or compound microscope, not a stereomicroscope.

Response: We have corrected this error. Thank you for pointing out this deficiency.

  1. add space after "version"

Response: We have added space. Thank you for pointing out this deficiency.

  1. Run-on sentence. There should be a semicolon after collection and "the" before 40.

Response: We have modified this sentence. Thank you for pointing out this deficiency.

  1. Decimal unneeded here.

Response: We have deleted the decimal. Thank you for pointing out this deficiency.

  1. "All" rather than whole????

Response: We have corrected this mistake. Thank you for pointing out this error.

Round 2

Reviewer 1 Report

This version is greatly improved beyond the original one. Thank you for all of this additional information and illustrations. Editorial recommendations are provided throughout the manuscript, mostly to improve the English grammar, especially in the newer portions. I look forward to seeing the published version.

Author Response

Dear Reviewer,

Thank you so much for the time and effort that you have put into reviewing our manuscript. We are particularly grateful to you for your careful editing of the language. Every one of your suggestions has been addressed. We have listed the changes to our manuscript below.

Looking forward to hearing from you soon.

Best regards

Sincerely yours

Haoming Zang, Beixin Wang, and Changhai Sun on behalf of all co-authors

The changes to our manuscript

  1. We have added information about Limnephilus tricalcaratus (Mosely, 1936) from China (Tibet) in our manuscript.

  1. We have counted Limnephilus kerekes Oláh, 2019; Limnephilus cheresnevi Nimmo, 1995; Limnephilus pallens (Banks, 1920); Limnephilus maghrebensis Mey & Oláh, 2019; Limnephilus oblos Oláh, 2019; Limnephilus tricalcaratus (Mosely, 1936), and Limnephilus barbagaensis Malicky, Sekhi & Lounaci 2019 into the 194 extant Limnephilus species in our manuscript.

  1. We have added 5 references to our manuscript.

  1. We have improved the English grammar according to your advice.

  1. We have used the figure label format required by Insects. (i.e. (Figure 2C–E, and Figure 6A–F).) Thank you for pointing out this deficiency.

  1. We have begun the key couplet with the name of the major structure in our manuscript.

  1. We have replaced all weak diagnostic characters in our manuscript.
